

# Factors Influencing Porosity in Planktonic Foraminifera

Janet E. Burke[1], Willem Renema[2], Michael J. Henehan[1,3], Leanne E. Elder[1], Catherine V. Davis[4], Amy E. Maas[5], Gavin L. Foster[6], Ralf Schiebel[7], Pincelli M. Hull[1]

5    [1] Department of Geology and Geophysics, Yale University, 210 Whitney Avenue, New Haven, CT 06511
[2] Naturalis Biodiversity Center, 2300 RA Leiden, The Netherlands
[3] GFZ German Research Centre for Geosciences, Telegrafenberg, 14473, Potsdam, Germany
[4] School of Earth, Oceans, and the Environment, University of South Carolina, 701 Sumter Street, EWS 617, Columbia, SC 29208
10  [5] Bermuda Institute of Ocean Sciences, 17 Biological Station, Ferry Reach, St. George's GE 01, Bermuda
[6] Ocean and Earth Science, University of Southampton, National Oceanography Centre, University Road, Southampton, SO17 1BJ, United Kingdom
[7] Max Planck Institut für Chemie, Hahn-Meitner-Weg 1, 55128 Mainz, Germany

Corresponding author: Janet E. Burke, janet.burke@yale.edu

**Abstract.** The clustering of mitochondria near pores in the test walls of foraminifera suggests that these perforations play a critical role in metabolic gas exchange. As such, pore measurements could provide a novel means of tracking changes in metabolic rate in the fossil record. However, in planktonic foraminifera, variation in pore size, density, and porosity have been variously attributed to environmental, biological, and taxonomic drivers, complicating such an interpretation. Here we examine the environmental, biological, and evolutionary determinants of porosity in 718 individuals representing 17 morphospecies of planktonic foraminifera from 6 core tops in the North Atlantic. Using random forest models, we find that porosity is primarily correlated to size and habitat temperature, two key factors in determining metabolic rates. In order to test if this correlation arose spuriously through the association of cryptic species with distinct biomes, we cultured *Globigerinoides ruber* in three different temperature conditions, and found that porosity increased with temperature. Crucially, these results show that porosity can be plastic: changing in response to environmental drivers within the lifetime of an individual foraminifer. This demonstrates the potential of porosity as a proxy for foraminiferal metabolic rates, with significance for interpreting geochemical data and the physiology of foraminifera in non-analog environments. It also highlights the importance of phenotypic plasticity (i.e., ecophenotypy) in accounting for some aspects of morphological variation in the modern and fossil record.

## 1 Introduction

Geochemical data from foraminiferal calcite often differs among species living in the same habitat due to biological factors collectively known as 'vital effects' (Erez, 1983; Spero et al. 1991; Ezard et al., 2015). Vital effects are often attributed, at least in part, to differences in metabolic processes such as respiration and photosynthesis (e.g. Wolf-Gladrow et al., 1999). Importantly though, these factors have not been directly measured in the vast majority of species leaving this idea largely untested (e.g. Ravelo & Fairbanks, 1995). A robust metabolic proxy could provide an independent constraint on the impact of vital effects on geochemical proxy signals such as $\delta^{13}C$ and $\delta^{11}B$ recorded in fossil foraminifera, thus impacting estimates of past atmospheric $CO_2$ concentrations (e.g. Anagnostou et al., 2016 ) and carbon cycling processes (e.g. Birch et al., 2016). Various aspects of foraminiferal test morphology have been observed to respond directly and measurably to metabolically-



relevant conditions in laboratory culture. For example, food quality and abundance can affect the terminal size of an adult foraminifer and the shape of its final chambers (Bé, 1982; Hemleben et al., 2012) and varying light levels have been related to changes in the size and shape of foraminiferal chambers in species that house photosynthetic symbionts (Bé, 1982; Spero,

1988; Bijma et al., 1992; Hemleben et al., 2012).

A particularly promising morphological characteristic that could provide insights into metabolic processes is porosity. Porosity is the total percent area of the test that is occupied by pores — small perforations in the tests of all planktonic foraminifera. The exact function of pores in foraminifera is not fully understood. Photosynthetic symbionts and mitochondria have been observed clustering near pores of benthic foraminifera (Hottinger & Dreher, 1974), and dissolved substances can be

absorbed through pores (Berthold, 1978). These observations suggest that pores may be involved in the physiological processes of osmoregulation and gas exchange. Porosity increases with the overall size of the test during ontogenetic development, potentially as a result of changes in depth ecology accompanying maturation, to accommodate increased movement of gas and solutes with increasing size, or to regulate buoyancy as the shell size increases (Bolli et al., 1994; Bé, 1968; Bé, et al 1973; Brummer et al., 1986; Marszalek et al., 1982; Huber et al., 1997; Schmidt et al., 2013).

Regardless of the exact function of pores, variation in porosity within and across species has frequently been attributed to environmental factors. A linear relationship exists between porosity and latitude, with higher porosities of >10% of the measured test wall associated with low latitudes and low porosities of <5% associated with high latitudes (Bé 1968; Frerichs et al., 1972). This pattern is commonly attributed to habitat temperature and has been used to track water masses during glacial-interglacial cycles in fossil and sub-fossil foraminiferal assemblages (Wiles, 1965; Bé, 1968; Frerichs et al., 1972; Bé &

Duplessy, 1976; Malmgren & Healy-Williams, 1978; Colombo & Cita, 1988; Fisher et al., 2003). Other environmental factors have also been hypothesized as drivers of morphological variation in porosity, including water density, salinity, oxygenation, and nitrogen concentration (Bé, 1968; Bé et al., 1973; Hottinger & Dreher, 1975; Berthold, 1978; Leutenegger & Hansen, 1979; Bé et al, 1980; Caron, 1987a,b; Hemleben et al., 2012; Bijma, et al., 1990; Moodley and Hess, 1992; Gupta & Machain-Castillo, 1993; Fisher, et al., 2003; Glock, 2011; Kuroyanagi, et al, 2013; Kuhnt, et al., 2014).

Pore variation across species and populations is also associated with evolutionary history. Pore size is the basis for a fundamental taxonomic division that distinguishes two major groups of planktonic foraminifera: the macroperforate (pores larger than 1μm in diameter) and microperforate (pores of 1μm or less) planktonic foraminifera (Bé et al., 1980; Kennett & Srinivasan, 1983; Qianyu & Radford, 1991). Within macroperforate planktonic foraminifera, there is a wide range of pore sizes and distribution patterns, some of which are characteristic of particular lineages. Globorotalid foraminifera, such as

*Globorotalia tumida* and *Globorotalia menardii*, can be distinguished from globigerinoid foraminifera like *Globigerinoides ruber* based on the shape, size and distribution of their pores (Bé et al., 1980). Porosity has also been used to distinguish between pseudo-cryptic species in modern foraminifera (Huber et al, 1997; Morard et al., 2009; Marshall et al., 2015; Weiner et al., 2015; Schiebel & Hemleben, 2017).

In summary, previous studies generally identify three different categories of factors influencing porosity: biological,

environmental, and phylogenetic. However, these factors are not independent of one another, and no previous study has attempted to detangle these various potential influences on porosity. Here we use core top samples from across the Atlantic Ocean to explore how porosity varies within and between populations, species, communities, size classes, and environments in order to identify the major determinants of porosity in modern macroperforate planktonic foraminifera. As an independent test



of the findings based on core tops, we also present cultured *Globigerinoides ruber* specimens grown in different temperature

conditions. These analyses are used together to consider the relationship between planktonic foraminiferal porosity and

metabolic processes including respiration and photosynthesis.

**2 Methods**

**2.1 Core Top Sample Selection and Processing**

Planktonic foraminifera from six Atlantic core-top localities spanning the major planktonic foraminifera biomes were sampled

from six sieve size fractions ranging from 150μm - 850μm (Figure 1, Table 1; biomes from Darling and Wade, 2008). At four

sites (KC 78, CH 82-21, VM 20-248, and EW 93-03-04; Figure 1), a random split of 50-100 individuals from each size fraction

was picked. At two additional sites, AII 60-10 and AII 42-15-14, target species were specifically picked to increase the

taxonomic and environmental range of our analyses (Table 1; Figure 1). Species were identified on the basis of the naming

conventions in Schiebel & Hemleben (2017). Specimens were mounted on microfossil slides and imaged at multiple focal

heights (z-stacks) from the spiral and umbilical side at a 10x magnification using a 5-megapixel Leica DFC450 digital camera

mounted on a Leica Microsystems DM6000M compound transmitted-light microscope with an automated x-y stepping stage

and drive focus. Umbilical views were used in the analysis of body size (see Supplemental Figures 1 and 2 and Supplemental

Discussion). Using *AutoMorph* (Hsiang et al., 2016; Hsiang et al, 2017), two-and three-dimensional shape and size information

was extracted from the z-stacked photographs of each individual, including surface area and volume (Supplemental Figure 1).

Two-dimensional measurements included cross-sectional area, major axis length, minor axis length, and perimeter length

(Supplemental Figure 1). Three-dimensional measurements included multiple estimates of volume and surface area using the

top (i.e., visible) half and a combination of visible top-halves with hypothetical backsides (see Hsiang et al., 2016;

Supplemental Figure 1).

 We include measures of both surface area and volume in our analysis due to their interactive effect on potential gas

exchange. Planktonic foraminifera with a flattened test shape (such as *Globorotalia menardii*) have a high surface area to

volume ratio, essentially maximizing the diffusive surface for their overall size. Conversely, spherical morphologies, like the

adult form of *Orbulina universa*, have the lowest possible surface-area to volume ratio for a given diameter, minimizing the

diffusive surface for their overall size. We focused on top-half estimates for this study because they are directly measured and

correlated with other estimates of surface area and volume (see Supplemental Figure 3). We were also interested in elliptical

estimates, as it has been suggested that, in vivo, spines and/or pseudopods would extend radially, making elliptical estimates

more representative of where respiration and photosynthesis take place (Zeebe et al., 1999). Elliptical estimates of surface area

and volume were calculated using height, length and width measurements assuming an elliptical solid. Because the two

measurements (top-half and elliptical) potentially represent different diffusive states that may be experienced by the living

organism, both were considered for the final analysis.

 After whole-specimen imaging, tests were dissected to remove the final and penultimate chamber and expose its inner wall

for porosity measurements (Figure 2). We quantified porosity from the inner wall of the penultimate chamber in order to avoid

known irregularities in the porosity of the final chambers (Bé et al, 1980; Constandache et al., 2017). In *Orbulina universa*, the



only exception, we measured the final chamber, as preceding chambers are typically dissolved in sedimentary remains of this species. Chamber fragments were then mounted on scanning electron microscope pins, coated in gold or platinum and carbon, and imaged in a scanning electron microscope (SEM) at a magnification of 300-600x to obtain the widest views of the inner chamber wall that were undistorted by the curvature of the chamber (Figure 2). SEM images were processed in ImageJ

(Schneider et al., 2012) to select an undistorted section of the chamber wall. The cropped image was converted to black and white and analyzed for the percent area occupied by pores (i.e., relative proportion of black pixels), average pore size, and total pore number. The total cropped area was used to convert pore number into a pore density estimate (i.e., number of pores/area). Images were cleaned if necessary to prevent debris from obscuring the pore measurements (Figure 2). Light photographic, SEM, and processed ImageJ images are provided through the Yale Peabody Museum collections portal

(http://collections.peabody.yale.edu/search/), using the Yale Peabody catalog numbers provided in Supplemental Table 1. Supplemental Tables 2-4 include all measurements collected for this study.

**2.2 Explanatory Variables**

We tested two-dimensional area, major axis length, top-half surface area, top-half volume, elliptical estimate surface area, elliptical estimate volume, sea surface temperature (SST), latitude, ambient temperature and oxygen concentration at habitat depth, and morphotype for their effect on porosity. Depth habitats were determined based on estimates from Schiebel & Hemleben (2017) and are given in Supplemental Table 7. Annual average sea surface temperature (SST, using temperature data from World Ocean Atlas for 10 meters depth), ambient temperature and oxygenation at depth habitat of each species were

obtained from World Ocean Atlas 2013 database (Locarnini et al., 2013 for temperature; Garcia et al., 2013 for oxygen) for each site and species (Supplemental Table 7). Morphogroups were globigerinid, globigerinoid, globorotalid and globoquadrinid as per Bé (1968)(Table 1).

**2.3 Cultured Samples**

Specimens of *Globigerinoides ruber* were cultured under controlled temperature conditions at the Bermuda Institute of Ocean Sciences in St. Georges, Bermuda in September 2016 in order to quantify the response of individual foraminiferal porosity to temperature. Specimens were live-caught 15-20 km off the coast of St. Georges, Bermuda (between 32.35012°N and 32.35942N, -64.59673°W and -64.68807°W) from the top 15 meters of the water column using a 150μm mesh Reeve net. All specimens were in the adult life stage at the time of the experiments. Specimens were picked from the towed material and

placed in recovery baths at 25°C until they showed signs of good health (spines, streaming cytoplasm, presence of symbionts, successful feeding) at which time they were moved to isolated culture jars and placed in a water bath held at a treatment temperature of 23°C, 25°C or 28°C. Both temperature and pH of the treatment water was monitored and kept stable throughout the experiments. Culture vial oxygen concentrations were checked for all temperature treatments with an oxygen optode attached to a Pyroscience FireSting optical oxygen meter to assure that concentrations did not fall below half saturation.

Specimens were fed single *Artemia* spp. nauplii and measured every other day to document growth. Specimens were kept in culture until they underwent gametogenesis or died (identified by the loss of cytoplasm within the test).



Specimens that accumulated 1 or more chambers in culture were imaged at a voxel size of 0.5-0.85μm using a Zeiss Xradia microXCT 400 at the University of Texas at Austin and a Zeiss Xradia 520 Versa micro-CT at Naturalis Biodiversity Center in Leiden, The Netherlands. Scanned specimens were reconstructed and extracted for inner wall porosity in VG StudioMax 3.0 using clipping planes and the ImageJ procedure explained above (Figure 2). In order to capture pre- and post-culture pore measurements for comparison pores were measured on the final 3-4 chambers.

**2.4 Statistical methods**

Random forest models were used to build predictive models and identify the major determinants for each pore characteristic (porosity, pore density, and pore size) using the *rpart*, *randomForest*, and *party* packages in R. Random forest models are supervised learning procedures that work by identifying the variables with the most explanatory power from a suite of theoretical decision trees (500 in this case) constructed from random samples of the data and predictor variables (Evans et al., 2011). The strength of each predictor variable is assessed by the reduction in model fit when that variable is excluded. In other words, the higher the percentage of incremental mean standard error associated with the removal of a variable, the higher that variable is ranked in terms of importance. They are robust to colinearity, nonlinearity, and deviations from normality in the data. Random Forests are useful for data sets with some missing data, and are applicable in situations without strong a priori hypothesis (Cutler et al., 2007; Davidson et al., 2009; Boyer, 2010). Even so, the variable importance rankings output by the standard random forest algorithm can be misleading if several explanatory variables covary and if the variables are of different types. Here, the environmental variables are strongly covariant and the model contains more than one variable type (all continuous except for morphotype, which is categorical). To account for this and aid in interpretation of the rankings, an unbiased, conditional variable importance ranking method was incorporated via the *party* package in R, which disentangles the most important variable from the model (Strobl et al, 2008). This method examines whether a correlation between the response variable and a predictor is conditional on another variable proceeding it in the tree, thereby identifying the most influential variable and demoting others (Strobl et al, 2008).

**2.5 Testing for Phylogenetic Signal**

Porosity, pore density, and average pore size were examined for a phylogenetic signal by estimating Pagel's lambda using average porosity for each species and the Cenozoic planktonic foraminiferal phylogeny of Aze et al. (2011). Pagel's lambda is a test designed to identify statistically significant grouping of trait values in phylogenetic clades as compared to the random distribution expected in the absence of a phylogenetic signal (Pagel, 1999). Pore measurement values were normalized using model residuals from random forests run without morphotype. A matrix of the average residual pore values for each species was created and analyzed using the "phylosig" function in the *phytools* package in R. The tree was trimmed of all branches lacking pore data.

**3 Results**



1,278 foraminifera were picked, imaged and identified to the morphospecies level for this study (Supplemental Figure 1). 718
specimens representing 17 morphospecies were successfully extracted for both two- and three-dimensional size metrics (i.e.,
surface area and volume), and are the focus of the statistical analyses presented here. Of the 17 morphospecies, 7 species occur
in 3 or more localities and 10 occur in 2 or more localities, allowing us to examine variation within morphospecies across
environments.

### 3.1 Factors influencing porosity in core top samples


In the original exploratory analyses (Supplemental Figure 3), six different, highly correlated measurements of test size were
examined. Using all of them in the random forest models would be redundant, so we ran iterations of the models with three
different sets of size variables—two-dimensional area and major axis length, top-half surface area and volume, and elliptical
surface area and volume—and chose the set which produced the model which explained the most variance in the porosity data.
We found that measurements of elliptical or top half surface area paired with volume always produced better-fitting models
than the two-dimensional measurements. These metrics better account for the surface area and volume disparities between
different morphologies that is lost in two-dimensional measurements. The elliptical and top-half measurement sets performed
comparably, but the top-half set produced a slightly stronger model for the porosity data set, so we used those measurements in
all three models for consistency. Random forest models were then built with the following seven variables: Top-half surface
area, top-half volume, sea surface temperature, morphogroup, ambient oxygen concentration, ambient temperature, and
latitude.

      The random forest model for the porosity data set explained 75.5% of the data. The most important variable was top-half
surface area, which caused a 34.1% increase in error when omitted from the model, followed by top-half volume and sea
surface temperature (26.6% and 23.8% increase in error, respectively; Figure 3; Table 2). The conditional variable analysis also
identified surface area as the most important variable. The random forest model for pore size explained 81.5% of the variance in
the data and was the strongest model built for the three different measures of pores (i.e., porosity, pore size, and pore
density). For pore size, ambient temperature was the strongest predictor (15.9% increase in error when absent) followed by sea-
surface temperature and surface area (14.5% and 12.8% increase in error when omitted from the model; Figure 3). In contrast
to the random forest model, the conditional variable analysis identified sea surface temperature as the most important variable
in explaining pore size, followed by latitude and ambient temperature (Table 2). The random forest for pore density explained
71.81% of the variance in the data. Here, morphotype was the most important factor (resulting in a 15.3% increase in model
error if omitted), followed by ambient temperature (11.3%; Figure 3). However, the conditional variable analysis (which is not
biased toward factors as random forests are) identified sea surface temperature and ambient temperature as the most important
variables.
220       Pore variables were compared against each other to consider their covariance. Within the pore characteristics, more of the
variation in porosity is explained by variation in pore size ($r^2$=0.64) than by pore density ($r^2$=0.17, Figure 4). Pairwise
relationships among porosity, pore size, and pore density were often non-linear and clustered by morphogroup (Figure 4).
Although globigerinoid foraminifera have a similar range of overall porosities to other morphogroups, they have the widest
range in pore sizes, and a narrow range of consistently low pore densities. These patterns in pore density and size, and other



characteristics not measured in this study like pore shape and rim type, are what makes the pore structures of these morphogroups distinguishable (Bé, 1980).

Model residuals for all three porosity variables were analyzed for phylogenetic signal using Pagel's Lambda. The lambda value was 0.245 for porosity (p-value=0.52), 1.09 for pore density (p-value=0.171), and >0.01 for pore size (p-value=1). This means that there was no significant phylogenetic signal detected for any of the three pore characteristics at 95% confidence

level. Even so, the lambda value of 1.09 for pore density indicates the presence of a phylogenetic signal at an 80% confidence level for pore density.

### 3.2 Temperature effect on cultured *Globigerinoides ruber*

The temperature experiments resulted in statistically significant differences in terminal porosity (Figure 5, Supplemental Figure 5) and body size. Average terminal porosity in low, medium and high temperature were 4.37% (1 standard deviation [s.d.] = 0.88%), 8.21% (1 s. d. = 1.33%), and 11.49% (1 s.d. = 0.905%). The groups were all statistically different according to a one-way ANOVA (F=57.1, p-value <0.01) and a pairwise Tukey's HSD post-hoc test (p<0.001 in all pairwise comparisons). Measurements of pre- and post-culture porosity from CT scans show a trend toward the treatment-average porosity as

chambers are accumulated (Supplemental Figure 5). In the high temperature treatment, pre-culture chambers all have porosities below 6%, but final cultured chamber porosities of above 10% by the end of the experiment. The specimens in the high temperature treatment also grew more chambers during their time in culture, with the high-temperature group accumulating an average of 0.45 chambers per day versus 0.38 and 0.24 for the low and medium temperature groups respectively. Average terminal size in low, medium and high temperature were 55334.$\mu m^2$ (1 s. d. = 17500.6 $\mu m^2$), 88430 $\mu m^2$ (1 s.d. = 32268.3 $\mu m^2$),

and 103394 $\mu m^2$ (1 s.d. = 36340.2 $\mu m^2$) respectively. The groups were all statistically different according to a one-way ANOVA (F= 93.573, p-value <0.001), but a pairwise Tukey's HSD post-hoc test showed that only the high and low temperature groups were significantly different (p=0.027 pairwise comparison). Pre-culture measurements of test area and porosity were not significantly different between treatments (Figure 5: F=1.184 and p>0.335 for test size, F=3.705 and p=0.062 for porosity), the high-temperature treatment foraminifera accumulated more chambers and achieved larger terminal sizes than

the low temperature group, and the size-normalized porosity was still significantly higher in the high-temperature group (Figure 5).

## 4 Discussion

Previous work on the porosity of planktonic foraminifera identified a number of environmental and biological correlates which often co-vary in time and space (Bé, 1968; Bé et al., 1976; Hottinger & Dreher, 1974; Berthold, 1978; Leutenegger & Hansen, 1979; Bé et al, 1980; Caron, 1987a,b; Hemleben et al., 2012; Bijma, et al., 1990; Moodley & Hess, 1992; Gupta & Machain-Castillo, 1992; Fisher, et al., 2003; Glock et al., 2011; Kuroyanagi et al., 2013). Our study builds on existing work by simultaneously investigating the three major types of drivers that may account for porosity: biology, environment, and

evolutionary history. Two key conclusions emerge from the models and experiments: that the main predictors on the porosity



of planktonic foraminifera are test size (specifically surface area) and temperature (Figure 3), and that both porosity and test size can be directly affected by changes in temperature during the life of an individual (Figure 5).

Both body size and temperature are known to have important effects on metabolism (Schmidt-Nielsen, 1984; Hochachka & Somero, 2002). Although there is variability among and within species, on average metabolic rate scales with body mass to the power of 3/4 in multicellular organisms (Kleiber 1961; Schmidt-Nielson, 1984; Brown et al., 2004), and 2/3 to 1 in protozoa (Caron et al., 1990; Agutter & Wheatley, 2004; Glazier, 2009). Overall size in planktonic foraminifera, similar to porosity, is smaller at high latitudes (Schmidt et al., 2013). Size variation, including changes in size throughout ontogeny, has been linked to variation in stable isotope values and the incorporation of trace metals into test calcite, possibly relating to variation in metabolic rate (e.g. Schmidt et al., 2008). Similarly, temperature has a powerful effect on metabolism that can be characterized by the respiratory $Q_{10}$ relationship—the factor by which an organism's respiration rate increases with a ten-degree increase in temperature. Estimates for the respiratory $Q_{10}$ of symbiont-bearing planktonic foraminifera (specifically *Globigerinoides ruber, Globigerinella siphonifera* and *Orbulina universa*) are approximately 3.18 (Lombard et al., 2009).

For single celled organisms like planktonic foraminifera, the metabolisms of large individuals are diffusion limited compared to small individuals, as volume increases to the third power, but surface area to the second. This is supported by our findings, which suggest that surface area was by far the most important factor in the porosity model (Fig. 3). To determine if porosity increases with temperature at the same rate as respiration, we calculated the change in porosity with a ten-degree change in sea-surface temperature (i.e. the $Q_{10}$ of porosity; Table 3; Supplementary Figure 6). To account for the effect of size on porosity, we first size-normalized the average porosity values by top-half surface area before calculating the $Q_{10}$ of porosity. We found an increase in porosity with temperature for nine of the ten species found at more than one site (i.e., all species in Table 3 except *Globorotalia inflata;* Supplementary Figure 6). For those species, size-normalized porosity increased by a $Q_{10}$ of porosity which varied from 1.3 to 2.4. This porosity $Q_{10}$ is close to the respiratory $Q_{10}$ of 3.18 and thus could relate to temperature-dependent variation in respiration rates.

Warmer water temperatures could lead to higher porosity for two reasons: warmer temperatures drive up metabolic rate and/or oxygen concentrations are lower in warmer water, necessitating higher rates of diffusion into the cell. For this reason, it was important to disentangle the effects of oxygen and temperature on porosity, and random forest models are specifically suited to dealing with such collinear variables. In all cases, oxygen was deemed less important than temperature and nearly all other variables considered. Our observations demonstrate that temperature is the underlying factor that drives the latitudinal trend in porosity observed by Bé (1968) at the species-level. Indeed, our results show a similar trend at the assemblage level and the morphospecies level: a decrease in average porosity with increasing latitude once normalized for size (i.e. top-half surface area, see Figure 6; Supplementary Figure 6). However, planktonic foraminifera species are known to inhabit characteristic biomes, and an alternative explanation for the apparent relationship between temperature and porosity could be that the change in porosity is driven by the turnover in species rather than temperature –in other words, by their shared evolutionary history. Three results argue against this alternative hypothesis. First, a phylogenetic signal was not found for porosity using Pagel's lambda. Second, morphotype (a coarse, categorical approximation for evolutionary relationship) explained relatively little of the variance in porosity in our random forest models and conditional variable analysis. Third, a two-way ANOVA to test for independent and interactive effects of species identity and temperature on the porosities of foraminifera, showed a much stronger effect of temperature ($F=594.42$, $p<0.001$), than the effect of species ($F=7.28$, $p<0.001$).



There was a significant interaction effect between the two factors, indicating that the two are not independent (F=7.3, p<0.001), and that species with higher porosities do occur at lower latitudes, and vice versa.

A second alternative explanation for the relationship between porosity and temperature is the presence of different pseudo-cryptic species across localities. Differences in porosity have been observed among genetic species within two morphospecies complexes: *Orbulina universa* and *Globigerinella siphonifera* (Huber et al., 1997; de Vargas et al., 1999; Morard et al., 2009; Morard et al., 2013; Marshall et al., 2015; Weiner et al., 2015). In fact, it is the sole characteristic by which two cryptic species of *Globigerinella siphonifera* can be identified in empty tests (Huber et al., 1997). In *Orbulina universa*, variation in areal

aperture density and placement distinguish among the three cryptic species, along with variation in wall-thickness in *Orbulina universa* (Morard et al., 2009; Marshall et al., 2015). We examined this, by culturing individuals of *Globigerinoides ruber* to test whether, and to what extent, porosity could vary based on environmental conditions at the time of chamber formation. We observed that individuals grown in the high temperature treatment became more porous, larger, and accumulated more chambers in culture as compared to those individuals grown in the low temperature treatment (similar to the findings of Bijma

et al., 1990) (Figure 5, Supplemental Figure 6). The average porosity of the high temperature group is approximately three times higher than that of the lower temperature group. Our culturing results indicate that porosity is highly plastic and varies rapidly in response to temperature changes in *Globigerinoides ruber*. Similarly, *Orbulina universa* cultured under different oxygen concentrations showed variation in areal aperture size as large as that observed across genetic species (Kuroyanagi et al., 2013).

Both culturing experiments point to the importance of environment in shaping the porosity of individuals, or ecophenotypy. Ecophenotypy in planktonic foraminifera has largely fallen out of favor as an explanation for variation in morphology, with the observations that ecophenotypes often align with different genetic complexes (Huber et al., 1997; de Vargas et al., 1999; de Vargas et al., 2001; Morard et al., 2009; Quillévéré et al., 2011; Morard et al., 2013; Marshall et al., 2015; Weiner et al., 2015). However, it is well-established that the expression of any phenotypic trait is a product of both its

genes and its environment (e.g. Visscher et al., 2008), with the heritability of a trait measuring the relative influence of genetics. In planktonic foraminifera, heritability has yet to be measured for any morphological trait, although it is likely to vary amongst traits as it does in all other organisms studied to date (Visscher et al., 2008). In this context, it is interesting to note that genetic-species of planktonic foraminifera are often found in distinct environments (i.e., different biomes or different depth habitats) (Huber et al., 1997; de Vargas et al., 2001; Darling & Wade, 2008; Morard et al., 2009; Quillévéré et al., 2011;

Morard et al., 2013; Morard et al., 2016). While evidence for high heritability of wall thickness and porosity is lacking, both porosity and wall thickness have been observed to vary with environmental conditions in culture and across environments gradients (this study; Colombo & Cita, 1980; Caron, 1987a-b; Bijma et al., 1990; Lea et al., 1999; Spero et al., 1997; Bijma et al., 1999; Russell et al., 2004; Lombard et al., 2009; Kuroyanagi et al., 2013; Spero et al., 2015; Henehan et al., 2017). This raises the interesting possibility that some of the morphological differences between different genetic species are driven

primarily by differences in the environment in which they occur, rather than by heritable genetic differences. While explanations of ecophenotypy have been dismissed in the past (Huber et al., 1997; Morard et al., 2009), our results suggest it should be seriously considered, at least for some traits like porosity, going forward.

Our results do show an evolutionary signal in some pore characteristics, but it is not the dominant factor in determining porosity. We find strong evidence for the importance of evolutionary history in determining pore density— one of the two



factors that together determine porosity (the other being pore size). Random forest models found morphotype to be the most important explanatory variable of pore density, although the conditional variance analysis attributed much of this explanatory power to a dependence on temperature (SST and ambient temperature). A Pagel's lambda of 1.09 for pore density on the model residuals likewise indicates a phylogenetic signal in the pore density data. Although this analysis was insignificant with alpha=0.1, we consider this finding important given the small sample size. For all three pore characteristics examined, pore

density, pore size, and the resultant porosity, morphotype does explain 12%-20% of the observed variation, so it is unsurprising that pore size has been such a useful trait for taxonomy. Similarly, the pairwise comparison of all three pore characteristics (Fig. 4) emphasizes the non-linear relationship between pore density and pore size, and the role of morphogroup in driving the bifurcating relationship between the two factors underlying porosity. However, when combined, the resulting porosity of an individual is more related to size and temperature, than it is to evolutionary history.


**5 Conclusion**

Test porosity in planktonic foraminifera from core top samples is primarily determined by size and temperature. These two factors are key determinants of respiration rate, and therefore suggest that porosity could be closely linked to metabolic rate –

likely through a role of porosity in allowing gas-exchange across the test wall. Experimental manipulations of *G. ruber* in cultures show that both test size and chamber porosity are sensitive to temperature, and that porosity is a plastic trait that responds to conditions experienced at the time of chamber formation. These results suggest that porosity has the potential to be a metabolic proxy that could aid in the interpretation of geochemical data and paleoecological reconstructions.


**Data Availability**

The data used in this study are available in the Supplemental Tables associated with this article.

**Competing Interests**

The authors declare that they have no conflict of interest.

**Author Contributions**


JEB and PMH designed the study and drafted the manuscript. JEB and WR conducted the CT scanning of cultured specimens. MJH, LEE, JEB, CVD, AEM, PMH, and GLF cultured foraminifera used in this study, and JEB, PMH, and RS identified the species. JEB extracted and analyzed the data. All coauthors contributed to writing the manuscript.


**Acknowledgements**

The authors of this manuscript would like to thank the following people for help in completing this work: Jessica Utrup of the Yale Peabody Museum for assistance in cataloging specimen images; Jessie Maisano at University of Texas and Dirk van der

Marel at Naturalis Biodiversity Center for help obtaining CT scans; Leocadio Blanco-Bercial, Samantha de Putron and the staff at BIOS for use of laboratory space and equipment; Kaylea Nelson at the Yale Center for Research Computing for assistance with the image analyses; Bruce Corliss from University of Rhode Island and Richard Norris from the Scripps Institute of Oceanography for the providing core-top materials; and Ellen Thomas and the members of the Hull Lab at Yale for feedback and comments on the manuscript. JEB was supported by the National Science Foundation Graduate Research Fellowship under

Grant No. DGE-1122492 and PMH by a Sloan Research Fellowship. Additional financial support for this work was provided



by the Cushman Foundation for Foraminiferal research, the Naturalis Biodiversity Center Martin Fellowship, the Bermuda Institute of Ocean Sciences Grants-in-Aid of Research program, and the Yale Peabody Museum.






**Table 1.** Locality and sieve size fraction for all core-top species sampled. Marker sizes correspond with sieve size fractions

(•=250-300μm, ○=300-425μm, ○=425-600μm, ○=600-710μm, ○=710-850μm).

| *Species* | Morphogroup | AII-42-15-14 | AII-60-10 | CH82 | EW9303 | KC78 | VM20 |
|---|---|---|---|---|---|---|---|
| *Globigerina bulloides* | Globigerina | | | ○ | •○ | | |
| *Globigerina falconensis* | Globigerina | | | | • | | |
| *Globigerinella siphonifera* | Globigerina | ○ | | | | •○ | •○ |
| *Globigerinoides conglobatus* | Globigerinoid | | ○ | ○ | | ○○○ | •○ |
| *Globigerinoides ruber* | Globigerinoid | ○ | • | ○ | | •○○ | •○ |
| *Globorotalia inflata* | Globorotalid | | • | ○○ | | | •○ |
| *Globorotalia crassaformis* | Globorotalid | | | ○○ | | | |
| *Globorotalia tumida* | Globorotalid | ○○ | | | | ○○ | |
| *Globorotalia hirsuta* | Globorotalid | | | ○○ | | | •○ |
| *Globorotalia menardii* | Globorotalid | | | | | ○○○○ | |
| *Neogloboquadrina dutertrei* | Globoquadrinid | | ○ | | | •○ | |
| *Neogloboquadrina incompta* | Globoquadrinid | | | | • | | |
| *Orbulina universa* | Globigerinoid | ○ | ○ | | | ○○○○ | |
| *Pulleniatina obliquiloculata* | Globoquadrinid | | | | | ○○ | |
| *Sphaeroidinella dehiscens* | Globigerinoid | | | | | ○○ | |
| *Globigerinoides sacculifer* | Globigerinoid | ○ | ○○ | | | •○○○○ | •○ |
| *Globorotalia truncatulinoides* | Globorotalid | ○○ | | ○○ | | | •○ |






**Table 2.** Variable importance rankings from random forest models and conditional variable importance analysis. RF variable importance rankings are based on the percent increase in error when the variable is removed from the model.


|  | RF Variable Importance | | | Conditional Variable Importance | | |
|---|---|---|---|---|---|---|
|  | Porosity | PD | AvP | Porosity | Pore Density | Pore Size |
| Surface Area | 34.98 | 11.54 | 10.39 | 0.532 | 0.033 | 0.015 |
| Volume | 21.40 | 1.95 | 8.29 | 0.138 | 0.012 | 0.022 |
| Sea Surface Temperature | 19.39 | 8.81 | 13.91 | 0.133 | 0.394 | 0.503 |
| Oxygen | 17.15 | 13.23 | 8.96 | 0.028 | 0.073 | 0.010 |
| Morphotype | 15.87 | 16.30 | 11.39 | 0.041 | 0.116 | 0.086 |
| Latitude | 15.68 | 9.54 | 11.31 | 0.074 | 0.160 | 0.201 |
| Habitat Temperature | 15.14 | 12.74 | 17.20 | 0.054 | 0.212 | 0.162 |



**Table 3.** Magnitude of porosity increase with a ten-degree temperature increase, as inferred from regressions of average size-normalized porosity and sea-surface temperature for core-top species that occurred at more than 2 localities. The size-normalized porosity average at ten degrees and twenty degrees is listed, along with the factor by which porosity increases over this interval are shown. See Supplemental Figure 6 for plots.

| *Species* | 10°C | 20°C | ΔPorosity | 95% Confidence Interval (+/-) |
|---|---|---|---|---|
| *Globigerinella siphonifera* | -0.171 | -0.098 | 1.745 | 0.011 |
| *Orbulina universa* | -0.25 | -0.104 | 2.403 | 0.022 |
| *Globigerinoides conglobatus* | -0.176 | -0.064 | 2.750 | 0.011 |
| *Globigerinoides ruber* | -0.122 | -0.092 | 1.326 | 0.005 |
| *Neogloboquadrina dutertrei* | -0.1648 | -0.0726 | 2.270 | 0.005 |
| *Trilobatus sacculifer* | -0.21 | -0.097 | 2.165 | 0.008 |
| *Truncorotalia truncatulinoides* | -0.091 | -0.071 | 1.282 | 0.012 |
| *Globoconella inflata* | -0.003 | 0.027 | -0.111 | 0.003 |



**Figure Captions**

Figure 1. Map of core-top sample localities (modified from Darling & Wade, 2008): a) EW 9303-04: 64.71°N, -28.91°E, Sub-Polar; b) CH 82-21: 43.288°N, -29.83°E, Transitional; c) VM 20-248: 33.5°N, -64.4°E, Sub-Tropical/Tropical; d) AII 42-15-14: 19.567°N, -44.95°E, Tropical; e) KC 78: 5.267°N, -44.133°E, Tropical; f) AII 60-10: -29.6°N, -34.667°E, Sub-Tropical.

Figure 2. Workflow diagrams for porosity and CT scan analyses. Pore characteristics for this study were measured on the internal test wall from SEM images of dissected foraminifera (pathway illustrated on the left side) or from CT scans (as shown on the right side). The method for extracting volume and surface area measurements is also shown on the far right.

Figure 3. Variable importance plots for the random forest models for each pore characteristic. Importance rankings are based on the increase in error produced when the variable in removed (% incremental mean squared error). Marker size refers to the ranking in the conditional variable importance analyses, with the largest markers denoting the most important variables.

Figure 4. Scatter plots of pore variables (with results of pairwise linear regressions) to visualize the relationship between pore variables.

Figure 5. Body size and porosity of cultured foraminifera grouped by treatment temperature for (a) the total change in area (major axis times minor axis), and (b) size-normalized terminal porosities.

Figure 6. Distribution of size-normalized porosity (%) values in each locality, arranged by latitude from lowest to highest. Grey boxes are samples from which a random split was taken, white boxes were picked for specific species.





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





**Figure 1.**

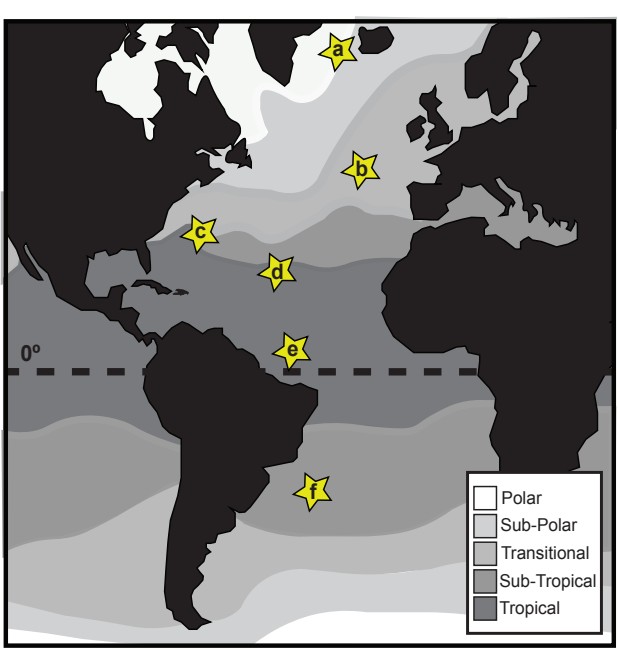





**Figure 2.**

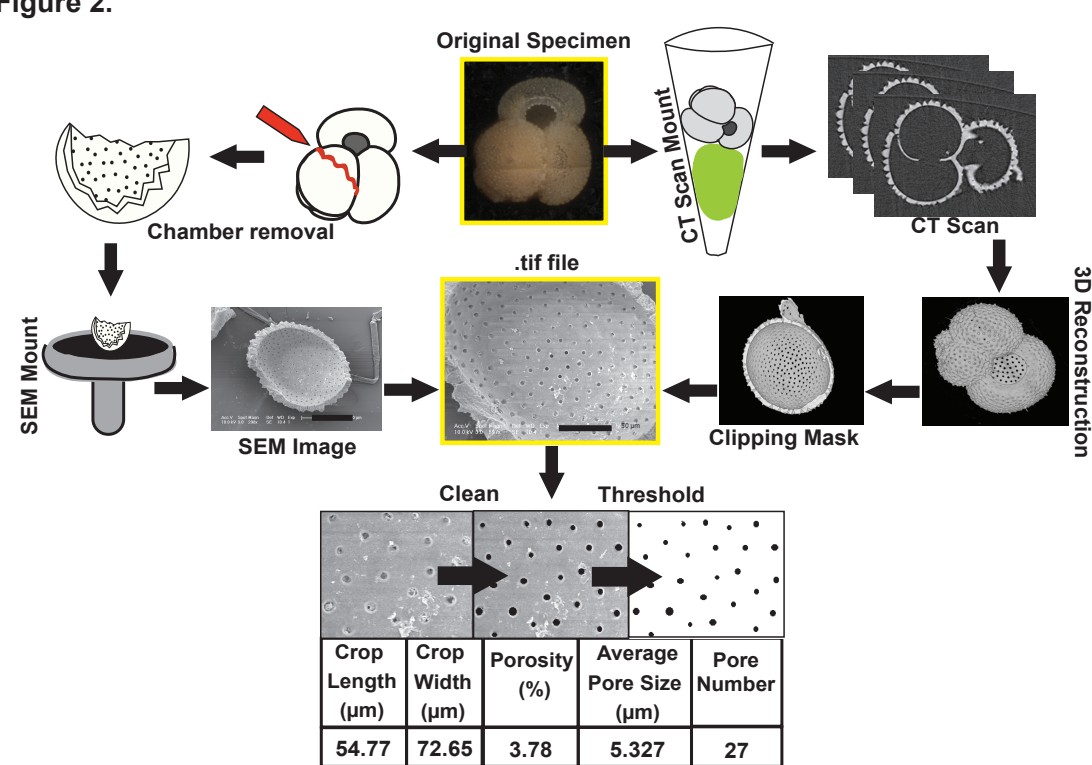





**Figure 3.**

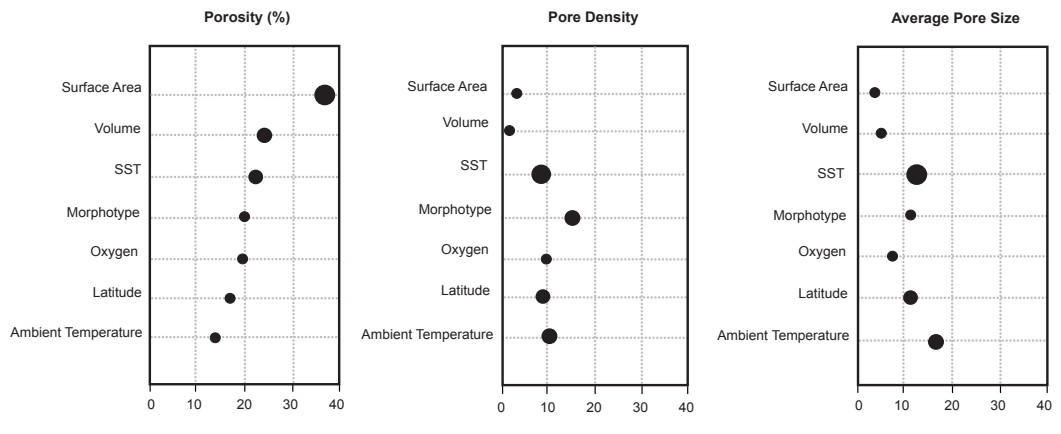

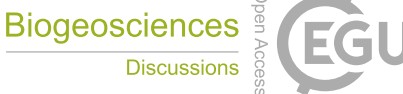




**Figure 4.**

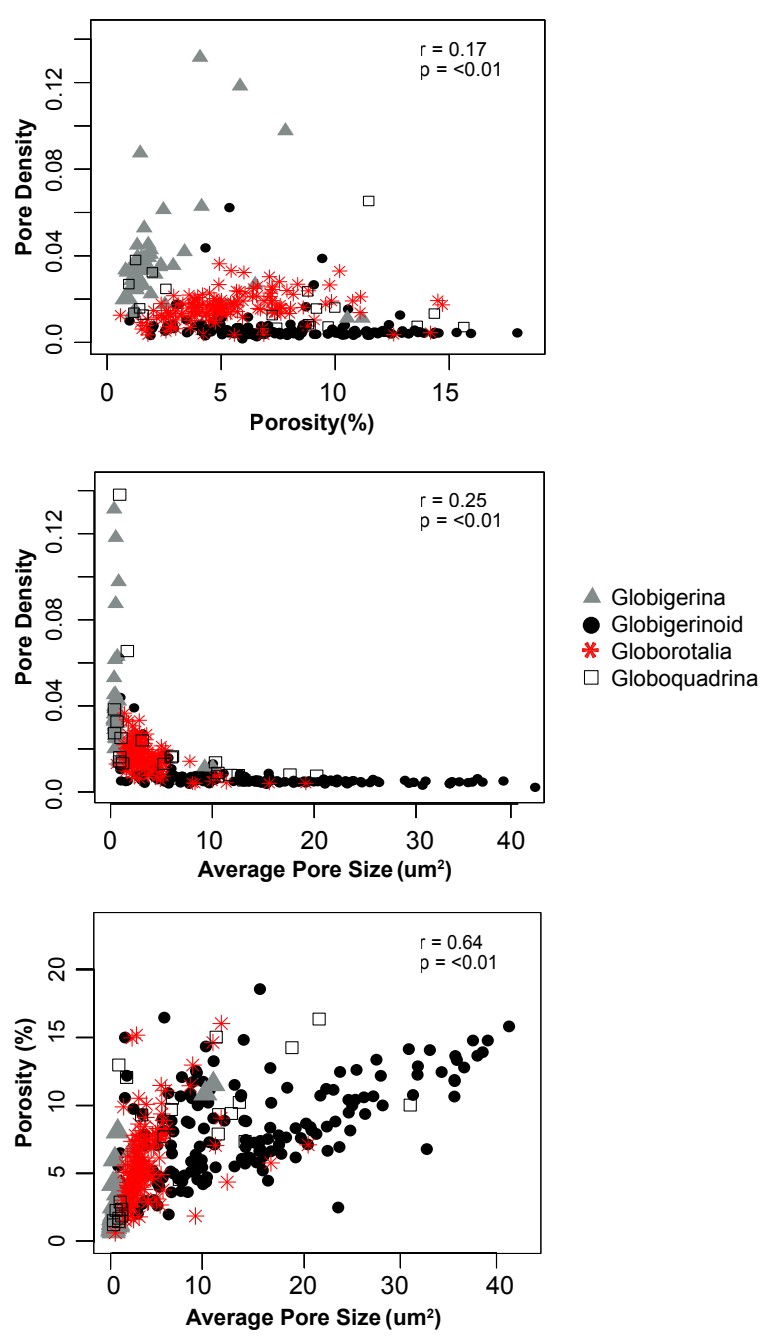



**Figure 5.**

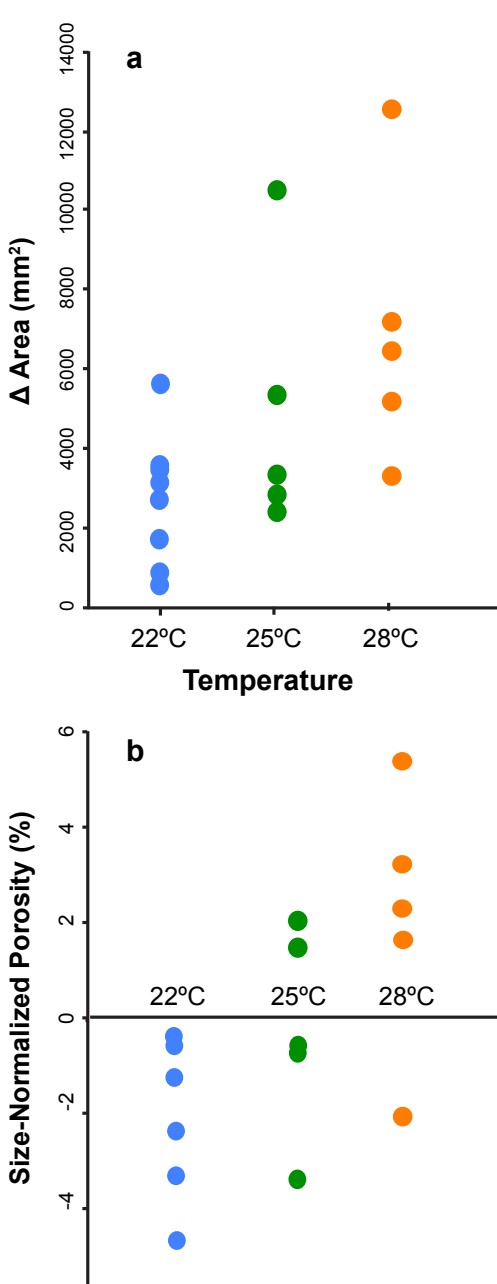





**Figure 6.**

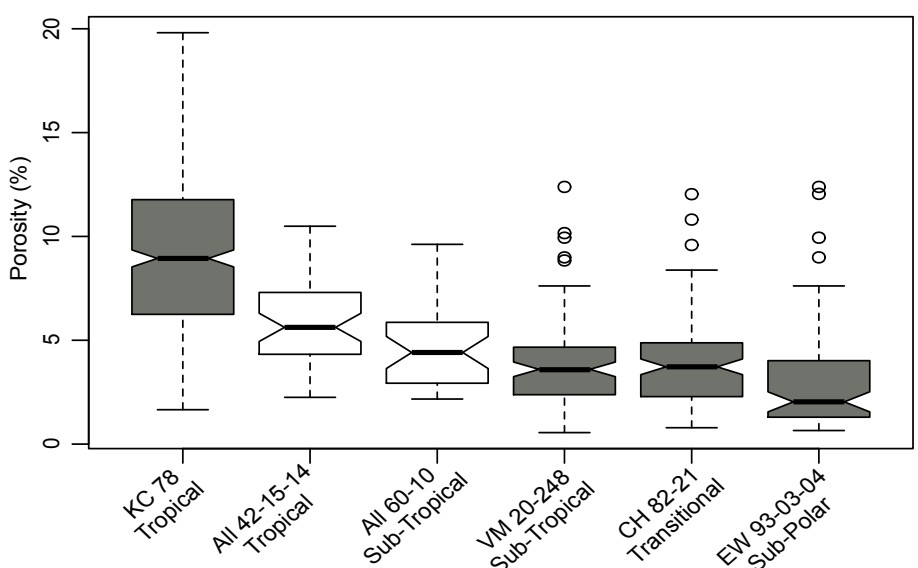