# Peer review of "Factors Influencing Porosity in Planktonic Foraminifera"

_Biogeosciences, 2018_

## Referee Comment (RC1) · A. Rathburn (Referee) · 19 Jun 2018

General Comments

This paper represents an important contribution to our understanding of the morphology, taxonomy, physiology and biogeochemistry of planktonic foraminifera. A combination of core top analyses and culturing techniques yielded an effective means to evaluate environmental and genetic influences on pore characteristics of a large number of specimens. The very interesting results from this study provide information that can be applied to the microfossil record of paleoceanographic change.

The paper is clear, well written, novel and timely.

Specific Comments

[Figure]

The term "porosity" has different meanings in geoscience. I think it might be best to use "surface area of pores" at least in the abstract, if not the title, to make it clearer what you are referring to.

What role does food availability play in the size and metabolism of planktonic foraminifera?

Technical Corrections

373 Should be "...Scripps Institution..."

---

## Referee Comment (RC2) · Anonymous Referee #2 · 10 Jul 2018

General comments

The manuscript entitled "Factors Influencing Porosity in Planktonic Foraminifera" by Burke et al. examined determinant factors of porosity of planktic foraminifers from core top samples using random forest models, considering environmental, biological, and taxonomic factors. They also conducted culture experiment to test the findings derived from the random forest models. They concluded that porosity is determined primarily by size and temperature that would be involved with metabolic rates. This study has fundamental importance on understanding the function of pores, application of porosity to reconstruct paleoecology, and interpretation of test geochemistry. The manuscript is well-written, and overall carefully discussed with statistical supports. However, there are some uncertainties that should be specified in the text, tables, and figures especially for the general terms like "size" and "porosity", and their units. In addition, discussion on temperature coefficient Q10 of porosity needs further consideration in terms of its calculation, interpretation, and the terminology as well. The paper would be more improved if the above points are considered.

Major points

1. Unit of "pore size"

In Figure 2, "average pore size" seems to have a unit of $\mu$m, so I thought it means pore diameter. However, when I carefully checked the dataset presented in the Supplementary Table 4 (I also downloaded some SEM images from YPM collections, and measured the pore diameter by myself), I found that the "pore size" values in the table seem to have a unit of $\mu$m2. Am I right? If so, I think "size" is not an appropriate term to represent an area of a pore (when we say test size, for instance, it usually indicates test diameter, not area). I also found that in Supplemental Figure 3b, "average pore size" is associated with a unit of $\mu$m, but in Figure 4 and in Supplementary Figure 6, "Pore size" is with $\mu$m2. Which is correct? Please clarify the definition of the parameter together with its unit. It is the same for "pore density". Perhaps it has a unit of "number $\mu$m-2", but please specify it as well.

2. Interpretation of Q10 of porosity

In the discussion part, Q10 of porosity is used to test if porosity increases with temperature at the same rate as respiration. The authors concluded that the Q10 of porosity ranging from 1.3 - 2.4 is close to that of respiration of 3.18 (Lombard et al. 2009), and it indicates the relation in respiration and porosity. In my understanding, however, these values can be said different. Since a Q10 value is a rate of change, 2-fold increase and 3-fold increase eventually cause a large difference. I agree that the porosity increases as temperature increases since the Q10 values are larger than 1 (except for G. inflata). However, the difference in Q10 of respiration and that of porosity is rather large. So, I would say the rate of respiration increase due to temperature rise is faster than the

increase of porosity. If the porosity and respiratory gas-exchange are related, it means that the gas-exchange becomes less efficient at a higher temperature (it might indicate that the porosity increase alone cannot meet the increasing respiratory gas-exchange). Maybe, for example, the presence of symbionts is involved with the efficient scavenging of respiratory gas... Anyway, please consider this point (i.e., the difference in Q10 of respiration and porosity) and add a bit more discussion in this part. In addition, according to the values shown in Table 3, the correct range of Q10 of porosity is "1.3 to 2.8", I suppose. Please reconfirm it.

3. Q10 calculation based on SST

I failed to understand why the authors chose SST to calculate Q10 of porosity instead of ambient temperature that directly affects physiological rates. SST can be an indicator of overall categorization of foraminiferal biomes, but it seems inappropriate to use it to calculate temperature sensitivity (i.e., Q10) of species. Especially, respiration of G. truncatulinoides that lives in deeper water mass won't be affected by SST. Would you please clarify this point, or is it possible to recalculate the Q10 of porosity based on the ambient temperature?

4. Use of the term Q10

In the first place, I wonder if it is appropriate to use the term Q10 for the case like porosity which is not a physiological or chemical reaction rate. In general, Q10 is used to show temperature sensitivity of biological (physiological) or chemical reaction rate. Q10 of porosity is understandable to me, but may not be a suitable terminology, simply because porosity is not a physiological rate. Please check the general usage of this terminology carefully.

5. "Size" of cultured specimens

The authors often mention on "body size" in Section 3.2 (e.g., L236, L420), but what this term indicates is not clear without very careful reading (I could understand that it

means the area, not the body mass or the test diameter, only after I reached L234). In the method part, please define the term. I recommend not to use "size" to indicate "area".

6. Size-normalized porosity

I failed to understand how the size-normalized porosity is calculated. Why the values with a unit of % have negative values (e.g., as represented in Figure 5b)? Would you please explain these values and how you calculated them in the method section or the supplementary text?

Minor points

L43, L45: Hemleben et al., 2012 —> Isn't it "Hemleben et al., 1989"? The book was firstly published in 1989, and later released as an e-book in 2012, I suppose.

L81: . . .including respiration and photosynthesis —> I did not see any discussion on porosity and photosynthesis in the text. If so, please delete "and photosynthesis". Meanwhile, I think it is good to add discussion on photosynthesis and porosity, if possible. Please see the abovementioned comment on Q10 of porosity.

L95-96: Supplemental Discussion —> I could not find "Supplemental Discussion" in supplementary materials. Perhaps you mean "Supplemental Text"?

L143: 32.35942N —> ° is missing.

L166: Random Forest —> Random forest

L245–247: "The groups were all statistically . . ..., but . . .." —> The wording sounds strange. Since one-way ANOVA is a method that evaluates whether the group means are drawn from populations with the same mean values or not, your one-way ANOVA result just shows there is a significant difference somewhere. It does not tell you that "the groups were all statistically different". Then, the post-hoc Tukey's HSD, a test to check where the difference exists, revealed that the significant difference exists between high- and low-temperature groups. So, the sentence should be "The groups were statistically different . . ., and a pairwise Tukey's . . ..".

L248: p>0.335 —> p=0.335?

L261: . . .test size (specifically surface area) —> How about just saying "surface area" since "test size" usually represents test diameter.

L624: Buma, J. —> Bijma, J.

Through the text: The number of decimal places is sometimes inconsistent among the same parameters (e.g., L216: 71.81% —> 71.8%, L228: p=0.52, 0.171, 1 —> 0.52, 0.71, 1.00(?), Table 3).

Through the text: "Supplemental Figure XX" or "Supplementary Figure XX"? Please use a consistent term.

Through the text: It seems that the term "porosity" is sometimes used in an expanded sense, not for the specific variable indicating the total percent area occupied by pores. In such cases, how about using "pore characteristics" instead? Otherwise, it is quite confusing.

Through the text: morphogroups or morphotype: In the text, both are used. If both represent the same categorization, please unify them to either one. In addition, the authors say "morphogroups were . . . as per Bé (1968)" in L136, but on the other hand, in the caption of Supplemental Figure 4, they say ". . .morphotype as described in Bé (1960)". Perhaps the latter should be Bé (1968)? Another concern relating to this is that morphogroups by Bé (1968) are based on test microstructure of species, including characteristics of perforation. Therefore, using this categorization to examine the effect of morphogroups on porosity seems to have a problem (maybe a kind of circular reasoning). Considering this point, the categorization of species should be solely based on, for example, genetic phylogeny (which is constructed independently from pore characteristics) in order to take into account for the evolutionary relationship. In fact, it will

not be a big problem because the categorization of morphogroups in this manuscript (i.e., globigeinoid, globigerinid, globoquadrinid, and globorotalid) are usually consistent to the other species categorization which is independent from pore characteristics.

Table 3: $\Delta$Porosity —> Does it mean Q10 of porosity? Please use the consistent term as appears in the text.

Table 3: Please use consistent genus names. If you use the naming convention in Schiebel and Hemleben (2017) as you declared in the text, Trilobatus should be Globigerinoides, Truncorotaria and Globoconella should be Globorotalia. It is the same for Supplemental Figure 4a, 4b, and 4c.

Figure 1: Please indicate longitude and latitude at least at the four end of the represented area.

Figure 4: The symbol for Globorotalia in the legend is not identical to the ones in the plot, strictly speaking. In addition, um2 should be $\mu$m2.

Figure 5, caption: Body size and porosity of…. —> Does "body size" mean "$\Delta$Area (mm2)" in Figure 5a? If so, I think the term is misleading, and needs to be corrected. In addition, more detailed explanation is needed in the caption as this is the only figure showing the results of cultured specimens except for supplementary figures.

Supplemental Figure 5: The colored bars are not easy to read especially in (b), and they are not so informative. I think it's okay without them. Alternatively, how about rearrange the panels to align each treatment group as a column (transpose columns and rows)? It will make it easy to compare different temperature treatments.

Supplemental Figure 6: What does the vertical axis mean? The caption says "size normalized porosity (%)", but in the figure, the axis is "Porosity residual".

---

## Author Comment (AC1) · 16 Jul 2018

We would like to thank Dr. Rathburn for his constructive and complimentary remarks on this manuscript and address the questions posed in the Reviewer Comment:

Comment: The term "porosity" has different meanings in geoscience. I think it might be best to use "surface area of pores" at least in the abstract, if not the title, to make it clearer what you are referring to.

Response: We agree that the general term porosity can be confusing, and will be more specific in the abstract and the title. We will change the title to "Factors Influencing Test Porosity in Planktonic Foraminifera" and introduce the term porosity as ". . . porosity (the total percentage of a test wall that is open pore space) . . ." on line 20 in the

abstract.

Comment: What role does food availability play in the size and metabolism of planktonic foraminifera?

Response: This is a good question, as food availability has been shown to effect terminal sizes and morphologies in laboratory culture (i.e. Lombard et al., 2011, Biogeosciences). There are two ways in which we might include the influence of food availability in our results: i) what is the influence of food availability on the individuals in the core top samples; ii) what is the influence of food availability on the culturing results? We begin with the second, as it helps inform our consideration of the first.

In culture, we fed individuals at what might be the upper end of their natural food intake: one large Artemia per day. It is notable in this context that the individuals cultured at near ambient temperatures ($25°$C), had culture porosities that were statistically indistinguishable from pre-culture values. This might suggest that food intake has a relatively modest effect on porosity.

However, perhaps we just happened to be feeding cultured individuals at comparable rates, amounts, and dietary compositions as their natural environment! We certainly did not directly test for an effect of food available on culture porosity.

Differences in major dietary factors (food type, symbiont status) may provide some additional insight into the influence of this food availability on porosity. In this context, it is notable that our morphogroups do roughly correspond with major food groups. For example, species in the globorotalid group are asymbiotic and species in the globigerinoid group have dinoflagellate symbionts. Also, we do find an effect of morphogroup on pore density, the number of pores in a unit of area.

However, this doesn't account for the difference in feeding frequency during an individual life span or in different spatially or temporally disparate populations of the same species. Given this, we will mention food availability as an important unknown and

target for future research in the discussion. At Line 309 we will insert a sentence that says: "Another environmental factor that may influence terminal sizes and metabolic function is the availability of food sources. Feeding frequency has been shown to influence terminal size and morphology (Bé, 1982; Hemleben et al., 1989), and can thus be expected to influence porosity as well. This factor is difficult to estimate for core top assemblages, but can be tested with simple culture experiments and subsequent imaging."

Comment: Technical Corrections 373 Should be ". . .Scripps Institution. . ."

Response: This correction will be made and reflected in future versions of the manuscript.

―――――――――――――――――

---

## Author Comment (AC2) · 7 Aug 2018

First, we would like to thank the Reviewer for their careful and constructive comments on the manuscript. The suggestions made will increase the clarity of the manuscript, and the concerns raised are important to address. We have the following responses to specific questions and comments:

Comment #1: Unit of "pore size" In Figure 2, "average pore size" seems to have a unit of  $\mu$ m, so I thought it means pore diameter. However, when I carefully checked the dataset presented in the Supplementary Table 4 (I also downloaded some SEM images from YPM collections, and measured the pore diameter by myself), I found that the "pore size" values in the table seem to have a unit of  $\mu$ m2. Am I right? If so, I

think "size" is not an appropriate term to represent an area of a pore (when we say test size, for instance, it usually indicates test diameter, not area). I also found that in Supplemental Figure 3b, "average pore size" is associated with a unit of  $\mu$ m, but in Figure 4 and in Supplementary Figure 6, "Pore size" is with  $\mu$ m2. Which is correct? Please clarify the definition of the parameter together with its unit. It is the same for "pore density". Perhaps it has a unit of "number  $\mu$ m-2", but please specify it as well.

Response #1: The reviewer is correct that the term "pore size" is indeed more accurately described as "average pore area" with a unit of "um2". All references to "pore size" and its units in the manuscript have been updated to correct this.

Comment #2: Interpretation of Q10 of porosity: In the discussion part, Q10 of porosity is used to test if porosity increases with temperature at the same rate as respiration. The authors concluded that the Q10 of porosity ranging from 1.3 - 2.4 is close to that of respiration of 3.18 (Lombard et al. 2009), and it indicates the relation in respiration and porosity. In my understanding, however, these values can be said different. Since a Q10 value is a rate of change, 2-fold increase and 3-fold increase eventually cause a large difference. I agree that the porosity increases as temperature increases since the Q10 values are larger than 1 (except for G. inflata). However, the difference in Q10 of respiration and that of porosity is rather large. So, I would say the rate of respiration increase due to temperature rise is faster than the increase of porosity. If the porosity and respiratory gas-exchange are related, it means that the gas-exchange becomes less efficient at a higher temperature (it might indicate that the porosity increase alone cannot meet the increasing respiratory gas-exchange). Maybe, for example, the presence of symbionts is involved with the efficient scavenging of respiratory gas. . . Anyway, please consider this point (i.e., the difference in Q10 of respiration and porosity) and add a bit more discussion in this part. In addition, according to the values shown in Table 3, the correct range of Q10 of porosity is "1.3 to 2.8", I suppose. Please reconfirm it.

Response #2: The Reviewer raises some important qualms with our discussion of res-

**BGD**
piratory Q10 and porosity. Although we were using the metric to aid comparisons, it is better to simply compare the respiratory Q10 obtained from laboratory study described in Lombard et al., 2009 to the porosity Q10, which we have for a more taxonomically and ecologically diverse set. The conclusion would be that if porosity and respiration were indeed linked to some extent, the level of variation in respiratory Q10 and porosity Q10 between the same species groups might look similar. Framing the discussion this way does not necessitate that respiration rate is the only driver of porosity or that the two have a 1:1 correlation, as the current discussion did. This allows for some rough assessment of the potential role of symbionts as well, an excellent suggestion by the reviewer, by comparing the porosity Q10 of symbiotic and asymbiotic species. A column listing the symbiont ecology of each species has been added to Table 3 (attached) and the section starting at Line 276, the section now reads:

"If porosity reflects metabolic rate, both should respond to temperature to a similar degree. To compare the temperature sensitivity of porosity with the respiratory and photosynthetic Q10 values (from Lombard et al., 2009), we calculated the change in size-normalized porosity with a ten-degree change in estimated ambient temperature (dubbed the Q10 of porosity; Table 3; Supplemental Figure 6). We found an increase in porosity with ambient temperature for six of the eight species found at more than one site (i.e., all species in Table 3 except Globorotalia inflata and Globorotalia truncatulinoides; Supplementary Figure 6). For those species, the Q10 of porosity varied from 1.3 to 2.3.

These porosity Q10 values are lower than the respiratory Q10 of 3.18 and the photosynthetic Q10 of 2.69 reported in Lombard et al., 2009. If porosity and respiratory gas-exchange are related, this means that either gas-exchange becomes less efficient at a higher temperature (suggesting that the porosity increase alone cannot meet the increasing respiratory gas-exchange demand), or that, since porosity is a physical property and not constrained by the same thermodynamic properties as the chemical reactions of photosynthesis and respiration, a 1 to 2 fold change is sufficient to reduce BGD
the diffusion limitation and meet the increase respiratory needs of the cells at higher temperatures. Alternatively, this discrepancy could be due to the fact that the measurements of Lombard et al. (2009) were taken from specimens exposed to sudden changes in temperature, which, as the authors noted, may result in higher sensitivity than that present in wild populations.

Furthermore, although we have hypothesized that respiratory demand for O2 is linked to pore size, it must be acknowledged that for symbiont bearing species, foraminifera metabolism is a complex interplay between photosynthesis and respiration. In some cases, where photosynthesis outpaces respiration, symbionts might provide O2 internally, reducing diffusion limitation. Alternatively the substrate demands (both O2 and CO2) and temperature sensitivity of the symbionts may be driving some of the observed porosity changes. On Table 3, species are sorted by Q10 of porosity from highest to lowest, with the symbiont ecologies of each group noted. Here, we can see that the species with the highest Q10 is a surface dweller with dinoflagellate symbionts (Globigerinoides conglobatus). The species with the lowest Q10 (Globorotalia truncatulinoides) is symbiont-barren with porosity that actually decreases with temperature. Additionally, the other species with a Q10 of less than one is Globorotalia inflata, a thermocline dweller with chrysophyte symbionts. These very low porosity Q10s might be due to the fact that the ambient temperatures are approximated from yearly averages of temperature at estimated depth habitats, or they may be a true reflection of a difference in porosity due to symbiont ecology. While Lombard et al., 2009 found that, after normalizing for cell size, the respiratory and photosynthetic Q10 of their specimens was consistent among the three species examined (Globigerinella siphonifera, Globigerinoides ruber, and Orbulina universa), what did differ between the species was the net photosynthesis to respiration ratio (P:R). Specifically, this ratio was much lower in the chrysophyte-bearing Globigerinella siphonifera than the dinoflagellate bearers Orbulina universa and Globigerinoides ruber. While we cannot conclude the extent of the relationship with our available data, the general trend of variation in Q10 of porosity roughly coinciding with symbiont ecology indicates that there may be some influence

**BGD**
of photosynthesis or photosynthesis to respiration ratio on porosity."

Additionally, the range of porosity in Line 281 will be corrected to read "1.3 to 2.3," the correct range.

Comment #3: Q10 calculation based on SST I failed to understand why the authors chose SST to calculate Q10 of porosity instead of ambient temperature that directly affects physiological rates. SST can be an indicator of overall categorization of foraminiferal biomes, but it seems inappropriate to use it to calculate temperature sensitivity (i.e., Q10) of species. Especially, respiration of G. truncatulinoides that lives in deeper water mass won't be affected by SST. Would you please clarify this point, or is it possible to recalculate the Q10 of porosity based on the ambient temperature?

Response #3: In response to this valid concern, especially as it relates to deep dwellers like G. truncatulinoides, I have recalculated the Q10 values for the updated manuscript using ambient temperature instead of sea surface temperature. I have also added a column to Table 3 for symbiont type to aid in discussion of the possible role of photosynthesis (shown in the response to comment #2).

Comment #4: Use of the term Q10 In the first place, I wonder if it is appropriate to use the term Q10 for the case like porosity which is not a physiological or chemical reaction rate. In general, Q10 is used to show temperature sensitivity of biological (physiological) or chemical reaction rate. Q10 of porosity is understandable to me, but may not be a suitable terminology, simply because porosity is not a physiological rate. Please check the general usage of this terminology carefully

Response #4: Temperature sensitivity of porosity is a major theme in this manuscript, and the Q10 term is a useful way to describe and compare this sensitivity. However, given that the exact physiological function of pores is unknown, the distinction between respiratory Q10 and porosity Q10 is communicated more explicitly in the updated version of the manuscript as shown in the response to Comment #2, specifically in the opening sentences:
"If porosity is reflecting metabolic rates, both should respond to temperature to a similar degree. To compare the temperature sensitivity of porosity with the respiratory and photosynthetic Q10 values (from Lombard et al., 2009), we calculated the change in porosity with a ten-degree change in estimated ambient temperature (dubbed the Q10 of porosity; Table 3; Supplemental Figure 6)."

Comment #5: "Size" of cultured specimens The authors often mention on "body size" in Section 3.2 (e.g., L236, L420), but what this term indicates is not clear without very careful reading (I could understand that it means the area, not the body mass or the test diameter, only after I reached L234). In the method part, please define the term. I recommend not to use "size" to indicate "area".

Response #5: "Size" and "body size" are used generally in the manuscript in reference to a number of different size-related parameters (cross-sectional area, surface area, volume, length, sieve size fraction). General terms have been replaced with specific terms in all references to size. For example, the final sentence in the discussion, formerly on Line 348, has been changed from:

"However, when combined, the resulting porosity of an individual is more related to size and temperature, than it is to evolutionary history."

To:

"However, when combined, the resulting porosity of an individual is more related to test surface area, test volume, and temperature, than it is to evolutionary history."

Also, the following sentence has been added to Line 101 of the Methods section:

"Size is an important factor in studies of planktonic foraminiferal ecology and biology, but it can refer to many different test parameters, like major axis length, aspect ratio, sieve size class, or three dimensional volume and surface area measurements. Here, we included two-dimensional area, major axis length, top-half surface area, top-half volume, elliptical estimate surface area, and elliptical estimate volume in the initial BGD
analyses to determine which set of size parameters was the most highly correlated with porosity. We include measurements of both surface area and . . ."

Comment #6: Size-normalized porosity I failed to understand how the size-normalized porosity is calculated. Why the values with a unit of % have negative values (e.g., as represented in Figure 5b)? Would you please explain these values and how you calculated them in the method section or the supplementary text?

Response #6: To control for differences in size, residuals from the porosity to surface area regression were used. These residuals are the values reported in the figures (Figure 5b; Supplemental Figures 4a-c and 6). In core top specimens the size variable is surface area, and in cultured specimens it is the two dimensional area (silhouette). The following sentence has been added to the methods section, starting at Line 111, to reflect this:

"Size-normalized porosity (i.e., the residuals from the porosity to surface area regression) was used in several analyses, where the aim was to explore the relationship between environmental variables and porosity regardless of the organism's size. To do this, residual porosity values from a regression of porosity and surface area (for core top specimens) or two-dimensional area (for cultured specimens) were used in lieu of direct porosity measurements."

Minor points

L43, L45: Hemleben et al., 2012 -> Isn't it "Hemleben et al., 1989"? The book was firstly published in 1989, and later released as an e-book in 2012, I suppose.

Response: This reference has been corrected to reflect the original release date of the text.

L81: . . . .including respiration and photosynthesis -> I did not see any discussion on porosity and photosynthesis in the text. If so, please delete "and photosynthesis". Meanwhile, I think it is good to add discussion on photosynthesis and porosity, if pos-

BGD
sible. Please see the abovementioned comment on Q10 of porosity.

Response: We agree with the suggestion that photosynthesis be discussed and have now incorporated symbiont ecology as it relates to Q10 of porosity into the discussion (in the response to Comment #2) and as a column in Table 3 (in the response to Comment #3).

L95-96: Supplemental Discussion -> I could not find "Supplemental Discussion" in supplementary materials. Perhaps you mean "Supplemental Text"?

Response: This line has been corrected to read "Supplemental Text" instead of "Supplemental discussion."

L143: 32.35942N -> [degree symbol] is missing.

Response: This line has been corrected.

L166: Random Forest -> Random forest

Response: This line has been corrected.

L245–247: "The groups were all statistically . . ..., but . . ..." -> The wording sounds strange. Since one-way ANOVA is a method that evaluates whether the group means are drawn from populations with the same mean values or not, your one-way ANOVA result just shows there is a significant difference somewhere. It does not tell you that "the groups were all statistically different". Then, the post-hoc Tukey's HSD, a test to check where the difference exists, revealed that the significant difference exists between high- and low-temperature groups. So, the sentence should be "The groups were statistically different . . ., and a pairwise Tukey's . . ..".

Response: This sentence has been changed as per the Reviewer's suggestion and now reads:

"The groups were all statistically different according to a one-way ANOVA (F= 93.57, p-value

low temperature groups were significantly different (p=0.03 pairwise comparison)."

L248: p>0.335 -> p=0.335? L261: . . .test size (specifically surface area) -> How about just saying "surface area" since "test size" usually represents test diameter.

Response: General references to "size", "test size", and "body size" have been replaced with the specific names for the measurements referenced.

L624: Buma, J. -> Bijma, J.

Response: This reference has been corrected.

Through the text: The number of decimal places is sometimes inconsistent among the same parameters (e.g., L216: 71.81% -> 71.8%, L228: p=0.52, 0.171, 1 -> 0.52, 0.71, 1.00(?), Table 3).

Response: The measurement values reported in the text have been carefully reviewed for consistency in significant figures.

Through the text: "Supplemental Figure XX" or "Supplementary Figure XX"? Please use a consistent term.

Response: All references to figures, tables and text has been changed to "Supplemental  $\ldots$ " in the next draft.

Through the text: It seems that the term "porosity" is sometimes used in an expanded sense, not for the specific variable indicating the total percent area occupied by pores. In such cases, how about using "pore characteristics" instead? Otherwise, it is quite confusing.

Response: The text has been reviewed carefully to correct all instances where porosity is used vaguely to refer to any pore variables, and these instances will be changed to "pore characteristics" as suggested.

Through the text: morphogroups or morphotype: In the text, both are used. If both
represent the same categorization, please unify them to either one. In addition, the authors say "morphogroups were . . . as per Bé (1968)" in L136, but on the other hand, in the caption of Supplemental Figure 4, they say ". . .morphotype as described in Bé (1960)". Perhaps the latter should be Bé (1968)? Another concern relating to this is that morphogroups by Bé (1968) are based on test microstructure of species, including characteristics of perforation. Therefore, using this categorization to examine the effect of morphogroups on porosity seems to have a problem (maybe a kind of circular reasoning). Considering this point, the categorization of species should be solely based on, for example, genetic phylogeny (which is constructed independently from pore characteristics) in order to take into account for the evolutionary relationship. In fact, it will not be a big problem because the categorization of morphogroups in this manuscript (i.e., globigeinoid, globigerinid, globoquadrinid, and globorotalid) are usually consistent to the other species categorization which is independent from pore characteristics.

Response: We have edited the manuscript to consistently refer to "morphogroups" and corrected the reference to Bé (1968). The exception is Line 190-194 the Results section where we describe identifying specimens to the morphospecies level.

Table 3:  $\Delta$ Porosity -> Does it mean Q10 of porosity? Please use the consistent term as appears in the text.

Response: This does mean Q10 of porosity and has been changed to reflect this.

Table 3: Please use consistent genus names. If you use the naming convention in Schiebel and Hemleben (2017) as you declared in the text, Trilobatus should be Globigerinoides, Truncorotaria and Globoconella should be Globorotalia. It is the same for Supplemental Figure 4a, 4b, and 4c.

This table has been corrected to be consistent with the taxonomy used in the rest of the manuscript.
Figure 1: Please indicate longitude and latitude at least at the four end of the represented area.

Response: I believe this comment is requesting that longitude and latitude markers be added to the far ends of the map. These have been added to the figure.

Figure 4: The symbol for Globorotalia in the legend is not identical to the ones in the plot, strictly speaking. In addition, um2 should be  $\mu$ m2.

Response: The symbol for Globorotalia in the legend has been un-bolded and the unit has been corrected in Figure 4.

Figure 5, caption: Body size and porosity of. . .. -> Does "body size" mean " $\Delta$ Area (mm2)" in Figure 5a? If so, I think the term is misleading, and needs to be corrected. In addition, more detailed explanation is needed in the caption as this is the only figure showing the results of cultured specimens except for supplementary figures.

Response: In this figure, "Body Size" is indeed two-dimensional area. The axis label now reflects this. The caption has been expanded to better describe the data as follows:

"Figure 5. Total test area and final chamber porosity of each cultured specimen of Globigerinoides ruber grouped by treatment temperature for (a) the total change in silhouette area before and after the experiment, and (b) size-normalized porosity of the final chamber. "

Supplemental Figure 5: The colored bars are not easy to read especially in (b), and they are not so informative. I think it's okay without them. Alternatively, how about rearrange the panels to align each treatment group as a column (transpose columns and rows)? It will make it easy to compare different temperature treatments.

Response: This figure has been re-drafted as per the Reviewer's suggestion.

Supplemental Figure 6: What does the vertical axis mean? The caption says "size
normalized porosity (%)", but in the figure, the axis is "Porosity residual".

Response: The vertical axis in this figure is the residual porosity value from a linear regression of porosity and surface area. I changed the axis label to "Size-normalized Porosity" and edited the caption to read as follows:

"Distribution of size-normalized porosity (%) values in each locality, arranged by latitude from lowest to highest. Porosity values shown are the residuals from a linear regression of surface area and porosity measurements."

References:

Bird et al., (2018). 16S rRNA gene metabarcoding and TEM reveals different ecological strategies within the genus Neogloboquadrina (planktonic foraminifer). PloS one, 13:1.

Ezard, T. H., et al. (2015). Environmental and biological controls on size-specific  $\delta$ 13C and  $\delta$ 18O in recent planktonic foraminifera. Paleoceanography, 30(3), 151-173.

Lombard, F., et al. (2009). Temperature effect on respiration and photosynthesis of the symbiont-bearing planktonic foraminifera Globigerinoides ruber, Orbulina universa, and Globigerinella siphonifera. Limnology and Oceanography, 54(1), 210-218.

Please also note the supplement to this comment: https://www.biogeosciences-discuss.net/bg-2018-222/bg-2018-222-AC2supplement.pdf

BGD

**Supplement:**

Table 3.

| Species | Porosity at 10°C | Porosity at 20°C | $Q_{10}$ Porosity | Symbiont Type |
|---|---|---|---|---|
| *Globigerinoides conglobatus* | -0.1757 | -0.0657 | 2.674 | Dinoflagellate[1] |
| *Neogloboquadrina dutertrei* | -0.1017 | -0.0397 | 2.562 | Pelagophtyes[2] |
| *Orbulina universa* | -0.195 | -0.084 | 2.321 | Dinoflagellate[1] |
| *Globigerinoides sacculifer* | -0.1698 | -0.0858 | 1.979 | Dinoflagellate[1] |
| *Globigerinella siphonifera* | -0.1625 | -0.0965 | 1.684 | Chrysophytes[1] |
| *Globigerinoides ruber* | -0.1104 | -0.0834 | 1.324 | Dinoflagellate[1] |
| *Globorotalia inflata* | -0.0628 | -0.0898 | 0.699 | Chrysophytes[1] |
| *Globorotalia truncatulinoides* | -0.0499 | -0.0869 | 0.574 | Asymbiotic[1] |

**[1] Ezard, T. H., et al. (2015). Environmental and biological controls on size‑specific δ13C and δ18O in recent planktonic foraminifera. *Paleoceanography*, *30*(3), 151-173.**
**[2] Bird et al., (2018). 16S rRNA gene metabarcoding and TEM reveals different ecological strategies within the genus Neogloboquadrina (planktonic foraminifer). *PloS one*, 13:1.**

---

## Author Response (AR1)

**Responses to All Reviewer Comments**
*Biogeosciences* Discussion Forum
"Factors Influencing Test Porosity in Planktonic Foraminifera"

Authors: Janet E. Burke, Willem Renema, Michael J. Henehan, Leanne E. Elder, Catherine V. Davis, Amy E. Maas, Gavin L. Foster, Ralf Schiebel, Pincelli M. Hull

**Reviewer #1: Anthony Rathburn**

Reviewer Comment: The term "porosity" has different meanings in geoscience. I think it might be best to use "surface area of pores" at least in the abstract, if not the title, to make it clearer what you are referring to.

Author Response: We agree that the general term porosity can be confusing, and will be more specific in the abstract and the title. We will change the title to "Factors Influencing Test Porosity in Planktonic Foraminifera" and introduce the term porosity as ". . . porosity (the total percentage of a test wall that is open pore space) . . ." on line 20 in the abstract.

RC: What role does food availability play in the size and metabolism of planktonic foraminifera?

AR: This is a good question, as food availability has been shown to effect terminal sizes and morphologies in laboratory culture (i.e. Hemleben et al., 1989). There are two ways in which we might worry about the influence of food availability in our results: i) what is the influence of food availability on the individuals in the core top samples; ii) what is the influence of food availability on the culturing results?  We begin with the second, as it helps inform our consideration of the first.

In culture, we fed individuals at what might be the upper end of their natural food intake: one large artemia per day. It is notable in this context that the individuals cultured at near ambient temperatures (25°C), had culture porosities that were statistically indistinguishable from pre-culture values.  This might suggest that food intake has a relatively modest effect on porosity.

However, perhaps we just happened to be feeding cultured individuals at comparable rates, amounts, and dietary compositions as their natural environment! We certainly did not directly test for an effect of food available on culture porosity.

Differences in major dietary factors (food type, symbiont status) may provide some additional insight into the influence of this food availability on porosity.  In this context, it is notable that our morphogroups do roughly correspond with major food groups. For example, species in the globorotalid group are asymbiotic and species in the globigerinoid group have dinoflagellate symbionts. Also, we do find an effect of morphogroup on pore density, the number of pores in a unit of area.

However, this doesn't account for the difference in feeding frequency during an individual life span or in different spatially or temporally disparate populations of the same species. Given this, we will mention food availability as an important unknown and target for future research in the discussion. At Line 309 we will insert a sentence that says:
 "Another environmental factor that may influence terminal sizes and metabolic function is the availability of food sources. Feeding frequency has been shown to influence terminal size and morphology (Bé, 1982; Hemleben et al., 1989), and may thus be expected to influence porosity as well. This factor is difficult to estimate for core top assemblages, but can be tested with simple culture experiments and subsequent imaging."

RC: Technical Corrections 373 Should be ". . .Scripps Institution. . ."

AR: This correction will be made and reflected in future versions of the manuscript.

**Reviewer #2: Anonymous**

Reviewer Comment: Unit of "pore size" In Figure 2, "average pore size" seems to have a unit of μm, so I thought it means pore diameter. However, when I carefully checked the dataset presented in the Supplementary Table 4 (I also downloaded some SEM images from YPM collections, and measured the pore diameter by myself), I found that the "pore size" values in the table seem to have a unit of μm2. Am I right? If so, I think "size" is not an appropriate term to represent an area of a pore (when we say test size, for instance, it usually indicates test diameter, not area). I also found that in Supplemental Figure 3b, "average pore size" is associated with a unit of μm, but in Figure 4 and in Supplementary Figure 6, "Pore size" is with μm2. Which is correct? Please clarify the definition of the parameter together with its unit. It is the same for "pore density". Perhaps it has a unit of "number μm-2", but please specify it as well.

Author Response: The reviewer is correct that the term "pore size" is indeed more accurately described as "average pore area" with a unit of "um$^2$". All references to "pore size" and its units in the manuscript have been updated to correct this.

RC: Interpretation of Q10 of porosity: In the discussion part, Q10 of porosity is used to test if porosity increases with temperature at the same rate as respiration. The authors concluded that the Q10 of porosity ranging from 1.3 - 2.4 is close to that of respiration of 3.18 (Lombard et al. 2009), and it indicates the relation in respiration and porosity. In my understanding, however, these values can be said different. Since a Q10 value is a rate of change, 2-fold increase and 3-fold increase eventually cause a large difference. I agree that the porosity increases as temperature increases since the Q10 values are larger than 1 (except for G. inflata). However, the difference in Q10 of respiration and that of porosity is rather large. So, I would say the rate of respiration increase due to temperature rise is faster than the increase of porosity. If the porosity and respiratory gas-exchange are related, it means that the gas-exchange becomes less efficient at a higher temperature (it might indicate that the porosity increase alone cannot meet the

increasing respiratory gas-exchange). Maybe, for example, the presence of symbionts is involved with the efficient scavenging of respiratory gas. . . Anyway, please consider this point (i.e., the difference in Q10 of respiration and porosity) and add a bit more discussion in this part. In addition, according to the values shown in Table 3, the correct range of Q10 of porosity is "1.3 to 2.8", I suppose. Please reconfirm it.

AR:  The Reviewer raises some important qualms with our discussion of respiratory $Q_{10}$ and porosity.  Although we were using the metric to aid comparisons, it is better to simply compare the respiratory $Q_{10}$ obtained from laboratory study described in Lombard et al., 2009 to the porosity $Q_{10}$, which we have for a more taxonomically and ecologically diverse set. The conclusion would be that if porosity and respiration were indeed linked to some extent, the level of variation in respiratory $Q_{10}$ and porosity $Q_{10}$ between the same species groups might look similar. Framing the discussion this way does not necessitate that respiration rate is the only driver of porosity or that the two have a 1:1 correlation, as the current discussion did. This allows for some rough assessment of the potential role of symbionts as well, an excellent suggestion by the reviewer, by comparing the porosity $Q_{10}$ of symbiotic and asymbiotic species. A column listing the symbiont ecology of each species has been added to Table 3 (below) and the section starting at Line 276, the section now reads:

*"If porosity reflects metabolic rate, both should respond to temperature to a similar degree. To compare the temperature sensitivity of porosity with the respiratory and photosynthetic $Q_{10}$ values (from Lombard et al., 2009), we calculated the change in size-normalized porosity with a ten-degree change in estimated ambient temperature (dubbed the $Q_{10}$ of porosity; Table 3; Supplemental Figure 6). We found an increase in porosity with ambient temperature for six of the eight species found at more than one site (i.e., all species in Table 3 except* Globorotalia inflata *and* Globorotalia truncatulinoides*; Supplementary Figure 6). For those species, the $Q_{10}$ of porosity varied from 1.3 to 2.3.*

*These porosity $Q_{10}$ values are lower than the respiratory $Q_{10}$ of 3.18 and the photosynthetic $Q_{10}$ of 2.69 reported in Lombard et al., 2009. If porosity and respiratory gas-exchange are related, this means that either gas-exchange becomes less efficient at a higher temperature (suggesting that the porosity increase alone cannot meet the increasing respiratory gas-exchange demand), or that, since porosity is a physical property and not constrained by the same thermodynamic properties as the chemical reactions of photosynthesis and respiration, a 1 to 2 fold change is sufficient to reduce the diffusion limitation and meet the increase respiratory needs of the cells at higher temperatures. Alternatively, this discrepancy could be due to the fact that the measurements of Lombard et al. (2009) were taken from specimens exposed to sudden changes in temperature, which, as the authors noted, may result in higher sensitivity than that present in wild populations.*

*Furthermore, although we have hypothesized that respiratory demand for O2 is linked to pore size, it must be acknowledged that for symbiont bearing species, foraminifera metabolism is a complex interplay between photosynthesis and respiration. In some cases, where photosynthesis outpaces respiration, symbionts might provide O2 internally, reducing diffusion limitation. Alternatively the substrate demands (both O2 and CO2) and temperature sensitivity of*

*the symbionts may be driving some of the observed porosity changes. On Table 3, species are sorted by $Q_{10}$ of porosity from highest to lowest, with the symbiont ecologies of each group noted. Here, we can see that the species with the highest $Q_{10}$ is a surface dweller with dinoflagellate symbionts (Globigerinoides conglobatus). The species with the lowest $Q_{10}$ (Globorotalia truncatulinoides) is symbiont-barren with porosity that actually decreases with temperature. Additionally, the other species with a $Q_{10}$ of less than one is Globorotalia inflata, a thermocline dweller with chrysophyte symbionts. These very low porosity Q10s might be due to the fact that the ambient temperatures are approximated from yearly averages of temperature at estimated depth habitats, or they may be a true reflection of a difference in porosity due to symbiont ecology. While Lombard et al., 2009 found that, after normalizing for cell size, the respiratory and photosynthetic $Q_{10}$ of their specimens was consistent among the three species examined (Globigerinella siphonifera, Globigerinoides ruber, and Orbulina universa), what did differ between the species was the net photosynthesis to respiration ratio (P:R). Specifically, this ratio was much lower in the chrysophyte-bearing Globigerinella siphonifera than the dinoflagellate bearers Orbulina universa and Globigerinoides ruber. While we cannot conclude the extent of the relationship with our available data, the general trend of variation in $Q_{10}$ of porosity roughly coinciding with symbiont ecology indicates that there may be some influence of photosynthesis or photosynthesis to respiration ratio on porosity."*

Additionally, the range of porosity in Line 281 will be corrected to read "1.3 to 2.3," the correct range.

| Species | Porosity at 10°C | Porosity at 20°C | $Q_{10}$ Porosity | Symbiont Type |
|---|---|---|---|---|
| *Globigerinoides conglobatus* | -0.1757 | -0.0657 | 2.674 | Dinoflagellate[1] |
| *Neogloboquadrina dutertrei* | -0.1017 | -0.0397 | 2.562 | Pelagophtyes[2] |
| *Orbulina universa* | -0.195 | -0.084 | 2.321 | Dinoflagellate[1] |
| *Globigerinoides sacculifer* | -0.1698 | -0.0858 | 1.979 | Dinoflagellate[1] |
| *Globigerinella siphonifera* | -0.1625 | -0.0965 | 1.684 | Chrysophytes[1] |
| *Globigerinoides ruber* | -0.1104 | -0.0834 | 1.324 | Dinoflagellate[1] |
| *Globorotalia inflata* | -0.0628 | -0.0898 | 0.699 | Chrysophytes[1] |
| *Globorotalia truncatulinoides* | -0.0499 | -0.0869 | 0.574 | Asymbiotic[1] |

[1] Ezard, T. H., et al. (2015). Environmental and biological controls on size-specific δ13C and δ18O in recent planktonic foraminifera. *Paleoceanography*, *30*(3), 151-173.
[2] Bird et al., (2018). 16S rRNA gene metabarcoding and TEM reveals different ecological strategies within the genus Neogloboquadrina (planktonic foraminifer). *PloS one*, 13:1.

RC: Q10 calculation based on SST I failed to understand why the authors chose SST to calculate Q10 of porosity instead of ambient temperature that directly affects physiological rates. SST can be an indicator of overall categorization of foraminiferal biomes, but it seems inappropriate to use it to calculate temperature sensitivity (i.e., Q10) of species. Especially, respiration of G. truncatulinoides that lives in deeper water mass won't be affected by SST. Would you please clarify this point, or is it possible to recalculate the Q10 of porosity based on the ambient temperature?

AR: In response to this valid concern, especially as it relates to deep dwellers like *G. truncatulinoides,* I have recalculated the $Q_{10}$ values for the updated manuscript using ambient temperature instead of sea surface temperature. I have also added a column to Table 3 for symbiont type to aid in discussion of the possible role of photosynthesis (shown in the response to comment #2).

RC: Use of the term Q10 In the first place, I wonder if it is appropriate to use the term Q10 for the case like porosity which is not a physiological or chemical reaction rate. In general, Q10 is used to show temperature sensitivity of biological (physiological) or chemical reaction rate. Q10 of porosity is understandable to me, but may not be a suitable terminology, simply because porosity is not a physiological rate. Please check the general usage of this terminology carefully

AR: Temperature sensitivity of porosity is a major theme in this manuscript, and the $Q_{10}$ term is a useful way to describe and compare this sensitivity. However, given that the exact physiological function of pores is unknown, the distinction between respiratory $Q_{10}$ and porosity $Q_{10}$ is communicated more explicitly in the updated version of the manuscript as shown in the response to Comment #2, specifically in the opening sentences:

*"If porosity is reflecting metabolic rates, both should respond to temperature to a similar degree. To compare the temperature sensitivity of porosity with the respiratory and photosynthetic $Q_{10}$ values (from Lombard et al., 2009), we calculated the change in porosity with a ten-degree change in estimated ambient temperature (dubbed the $Q_{10}$ of porosity; Table 3; Supplemental Figure 6)."*

RC: "Size" of cultured specimens:  The authors often mention on "body size" in Section 3.2 (e.g., L236, L420), but what this term indicates is not clear without very careful reading (I could understand that it means the area, not the body mass or the test diameter, only after I reached L234). In the method part, please define the term. I recommend not to use "size" to indicate "area".

AR: "Size" and "body size" are used generally in the manuscript in reference to a number of different size-related parameters (cross-sectional area, surface area, volume, length, sieve size fraction). General terms have been replaced with specific terms in all references to size. For example, the final sentence in the discussion, formerly on Line 348, has been changed from:

*"However, when combined, the resulting porosity of an individual is more related to size and temperature, than it is to evolutionary history."*

To:

*"However, when combined, the resulting porosity of an individual is more related to test surface area, test volume, and temperature, than it is to evolutionary history."*

Also, the following sentence has been added to Line 101 of the Methods section:

*"Size is an important factor in studies of planktonic foraminiferal ecology and biology, but it can refer to many different test parameters, like major axis length, aspect ratio, sieve size class, or three dimensional volume and surface area measurements. Here, we included two-dimensional area, major axis length, top-half surface area, top-half volume, elliptical estimate surface area, and elliptical estimate volume in the initial analyses to determine which set of size parameters was the most highly correlated with porosity. We include measurements of both surface area and . . ."*

RC: Size-normalized porosity I failed to understand how the size-normalized porosity is calculated. Why the values with a unit of % have negative values (e.g., as represented in Figure 5b)? Would you please explain these values and how you calculated them in the method section or the supplementary text?

AR: To control for differences in size, residuals from the porosity to surface area regression were used. These residuals are the values reported in the figures (Figure 5b; Supplemental Figures 4a-c and 6). In core top specimens the size variable is surface area, and in cultured specimens it is the two dimensional area (silhouette). The following sentence has been added to the methods section, starting at Line 111, to reflect this.

*"Size-normalized porosity (i.e., the residuals from the porosity to surface area regression) was used in several analyses, where the aim was to explore the relationship between environmental variables and porosity regardless of the organism's size. To do this, residual porosity values from a regression of porosity and surface area (for core top specimens) or two-dimensional area (for cultured specimens) were used in lieu of direct porosity measurements."*

RC: L43, L45: Hemleben et al., 2012 —> Isn't it "Hemleben et al., 1989"? The book was firstly published in 1989, and later released as an e-book in 2012, I suppose.

AR: This reference has been corrected to reflect the original release date of the text.

RC: L81: . . .including respiration and photosynthesis —> I did not see any discussion on porosity and photosynthesis in the text. If so, please delete "and photosynthesis". Meanwhile, I think it is good to add discussion on photosynthesis and porosity, if possible. Please see the abovementioned comment on Q10 of porosity.

AR: We agree with the suggestion that photosynthesis be discussed and have now incorporated symbiont ecology as it relates to Q10 of porosity into the discussion (in the response to Comment #2) and as a column in Table 3 (in the response to Comment #3).

RC: L95-96: Supplemental Discussion —> I could not find "Supplemental Discussion" in supplementary materials. Perhaps you mean "Supplemental Text"?

AR: This line has been corrected to read "Supplemental Text" instead of "Supplemental discussion."

RC: L143: 32.35942N —> ∘ is missing.

AR: This line has been corrected.

RC: L166: Random Forest —> Random forest

AR: This line has been corrected.

RC: L245–247: "The groups were all statistically . . ..., but . . ..." —> The wording sounds strange. Since one-way ANOVA is a method that evaluates whether the group means are drawn from populations with the same mean values or not, your one-way ANOVA result just shows there is a significant difference somewhere. It does not tell you that "the groups were all statistically different". Then, the post-hoc Tukey's HSD, a test to check where the difference exists, revealed that the significant difference exists between high- and low-temperature groups. So, the sentence should be "The groups were statistically different . . ., and a pairwise Tukey's . . .."".

AR: This sentence has been changed as per the Reviewer's suggestion and now reads:

*"The groups were all statistically different according to a one-way ANOVA (F= 93.57, p-value <0.001). A pairwise Tukey's HSD post-hoc test showed that only the high and low temperature groups were significantly different (p=0.03 pairwise comparison)."*

RC: L248: p>0.335 —> p=0.335? L261: . . .test size (specifically surface area) —> How about just saying "surface area" since "test size" usually represents test diameter.

AR: General references to "size", "test size", and "body size" have been replaced with the specific names for the measurements referenced.

RC: L624: Buma, J. —> Bijma, J.

AR: This reference has been corrected.

RC: Through the text: The number of decimal places is sometimes inconsistent among the same parameters (e.g., L216: 71.81% —> 71.8%, L228: p=0.52, 0.171, 1 —> 0.52, 0.71, 1.00(?), Table 3).

AR: The measurement values reported in the text have been carefully reviewed for consistency in significant figures.

RC: Through the text: "Supplemental Figure XX" or "Supplementary Figure XX"? Please use a consistent term.

AR: All references to figures, tables and text has been changed to "Supplemental …" in the next draft.

RC: Through the text: It seems that the term "porosity" is sometimes used in an expanded sense, not for the specific variable indicating the total percent area occupied by pores. In such cases, how about using "pore characteristics" instead? Otherwise, it is quite confusing.

JEB: The text has been reviewed carefully to correct all instances where porosity is used vaguely to refer to any pore variables, and these instances will be changed to "pore characteristics" as suggested.

RC: Through the text: morphogroups or morphotype: In the text, both are used. If both represent the same categorization, please unify them to either one. In addition, the authors say "morphogroups were . . . as per Bé (1968)" in L136, but on the other hand, in the caption of Supplemental 4, they say ". . .morphotype as described in Bé (1960)". Perhaps the latter should be Bé (1968)? Another concern relating to this is that morphogroups by Bé (1968) are based on test microstructure of species, including characteristics of perforation. Therefore, using this categorization to examine the effect of morphogroups on porosity seems to have a problem (maybe a kind of circular reasoning). Considering this point, the categorization of species should be solely based on, for example, genetic phylogeny (which is constructed independently from pore characteristics) in order to take into account for the evolutionary relationship. In fact, it will not be a big problem because the categorization of morphogroups in this manuscript (i.e., globigeinoid, globigerinid, globoquadrinid, and globorotalid) are usually consistent to the other species categorization which is independent from pore characteristics.

AR: We have edited the manuscript to consistently refer to "morphogroups" and corrected the reference to Bé (1968). The exception is Line 190-194 the Results section where we describe identifying specimens to the morphospecies level.

RC: Table 3: ΔPorosity —> Does it mean Q10 of porosity? Please use the consistent term as appears in the text.

AR: This does mean Q10 of porosity and has been changed to reflect this.

RC: Table 3: Please use consistent genus names. If you use the naming convention in Schiebel and Hemleben (2017) as you declared in the text, Trilobatus should be Globigerinoides, Truncorotaria and Globoconella should be Globorotalia. It is the same for Supplemental Figure 4a, 4b, and 4c.

AR: This table has been corrected to be consistent with the taxonomy used in the rest of the manuscript.

RC: Figure 1: Please indicate longitude and latitude at least at the four end of the represented area.

AR: I believe this comment is requesting that longitude and latitude markers be added to the far ends of the map. These have been added to the figure.

RC: Figure 4: The symbol for Globorotalia in the legend is not identical to the ones in the plot, strictly speaking. In addition, um2 should be µm2.

AR: The symbol for Globorotalia in the legend has been un-bolded and the unit has been corrected in Figure 4.

RC: Figure 5, caption: Body size and porosity of. . .. —> Does "body size" mean "ΔArea (mm2)" in Figure 5a? If so, I think the term is misleading, and needs to be corrected. In addition, more detailed explanation is needed in the caption as this is the only figure showing the results of cultured specimens except for supplementary figures.

AR: In this figure, "Body Size" is indeed two-dimensional area. The axis label now reflects this. The caption has been expanded to better describe the data as follows:

*"Figure 5. Total test area and final chamber porosity of each cultured specimen of Globigerinoides ruber grouped by treatment temperature for (a) the total change in silhouette area before and after the experiment, and (b) size-normalized porosity of the final chamber. "*

RC: Supplemental Figure 5: The colored bars are not easy to read especially in (b), and they are not so informative. I think it's okay without them. Alternatively, how about rearrange the panels to align each treatment group as a column (transpose columns and rows)? It will make it easy to compare different temperature treatments.

AR: This figure has been re-drafted as per the Reviewer's suggestion.

RC: Supplemental Figure 6: What does the vertical axis mean? The caption says "size normalized porosity (%)", but in the figure, the axis is "Porosity residual".

AR: The vertical axis in this figure is the residual porosity value from a linear regression of porosity and surface area. I changed the axis label to "Size-normalized Porosity" and edited the caption to read as follows:

*"Distribution of size-normalized porosity (%) values in each locality, arranged by latitude from lowest to highest. Porosity values shown are the residuals from a linear regression of surface area and porosity measurements."*

[revised manuscript text omitted]

Jana Burke 8/27/18 3:26 PM
Unknown

600

**Figure 3.**

[Figure]

**Figure 4.**

[Figure]

**Figure 5.**

[Figure]

**Figure 6.**

[Figure]